



# Variability in epilimnion depth estimations in lakes

Harriet L. Wilson[1], Ana I. Ayala[2], Ian D. Jones[3], Alec Rolston[4], Don Pierson[2], Elvira de Eyto[5], Hans-Peter Grossart[6], Marie-Elodie Perga[7], R. Iestyn Woolway[1], Eleanor Jennings[1]

[1]Center for Freshwater and Environmental Studies, Dundalk Institute of Technology, Dundalk, Ireland
[2]Department of Ecology and Genetics, Limnology, Uppsala University, Uppsala, Sweden
[3]Biological and Environmental Sciences, Faculty of Natural Sciences, University of Stirling, Stirling, UK
[4]An Fóram Uisce, National Water Forum, Ireland
[5]Marine Institute, Furnace, Newport, Co. Mayo, Ireland
[6]Institute for Biochemistry and Biology, Potsdam University, Potsdam, Germany
[7]University of Lausanne, Faculty of Geoscience and Environment, CH 1015 Lausanne, Switzerland

*Correspondence to*: Harriet L. Wilson (wilsonh@dkit.ie)

**Abstract.** The "epilimnion" is the surface layer of a lake typically characterised as well-mixed and is decoupled from the "metalimnion" due to a rapid change in density. The concept of the epilimnion, and more widely, the three-layered structure of a stratified lake, is fundamental in limnology and calculating the depth of the epilimnion is essential to understanding many physical and ecological lake processes. Despite the ubiquity of the term, however, there is no objective or generic approach for defining the epilimnion and a diverse number of approaches prevail in the literature. Given the increasing availability of water temperature and density profile data from lakes with a high spatio-temporal resolution, automated calculations, using such data, are particularly common, and have vast potential for use with evolving long-term, globally measured and modelled datasets. However, multi-site and multi-year studies, including those related to future climate impacts, require robust and automated approaches for epilimnion depth estimation. In this study, we undertook a comprehensive comparison of commonly used epilimnion depth estimation methods, using a combined 17 year dataset, with over 4700 daily temperature profiles from two European lakes. Overall, we found a very large degree of variability in the estimated epilimnion depth across all methods and thresholds investigated and for both lakes. These differences, manifest over high-frequency data, led to fundamentally different understandings of the epilimnion depth. In addition, estimations of the epilimnion depth were highly sensitive to small changes in the threshold value, complex water column structures and vertical data resolution. These results call into question the custom of arbitrary method selection, and the potential problems this may cause for studies interested in estimating the ecological processes occurring within the epilimnion, multi-lake comparisons or long-term time series analysis. We also identified important systematic differences between methods, which demonstrated how and why methods diverged. These results may provide rationale for future studies to select an appropriate epilimnion definition in light of their particular purpose and with awareness of the limitations of individual methods. While there is no prescribed rationale for selecting a particular method, the method which defined the epilimnion depth as the shallowest depth where the density was 0.1 kg m$^{-3}$ more than the surface density, was shown to be overall less problematic than the other methods.

## 1 Introduction

The "epilimnion depth", "mixed layer" or "top of the metalimnion" are common terms in limnology, typically referring to the deepest point of the surface layer of a stratified lake, characterised as quasi-uniform in terms of physical and biogeochemical properties, and overlying a layer of rapid vertical gradients. Incoming heat to a lake, received at the lake surface, expands water above 3.98 °C resulting in density stratification. Convective cooling at the surface and mechanical energy injected by the wind, drive vertical mixing (Wüest and Lorke, 2003). These competing surface fluxes result in a





warm, well mixed layer of water that interacts dynamically with the atmosphere (Monismith and MacIntyre, 2008). The vertical propagation of energy manifested at the lake surface is constrained by the sharp density gradients in the metalimnion, which act to decouple the epilimnion from the deep hypolimnion. As such, it has become foundational in limnology to consider a stratified lake as consisting of three well-defined layers, a turbulent epilimnion (diffusivity typically $10^{-5}$ - $10^{-2}$ m$^2$ s$^{-1}$), the stable metalimnion (5 x $10^{-8}$ - $10^{-6}$ m$^2$ s$^{-1}$) and the quiescent hypolimnion (3 x $10^{-6}$ - $10^{-4}$ m$^2$ s$^{-1}$) (Wüest and Lorke, 2009). The discretisation of these layers, however, is understood to be theoretical, since micro-profile studies show that the conditions within layers are not uniform and exact cut-offs between layers do not necessarily exist (Imberger, 1985, Jonas et al., 2003, Kraemer et al., 2020).

Quantifying the vertical extent of the epilimnion is crucial for understanding many of the physical, chemical and biological processes in lakes. Although the epilimnion is differentiated from the typically shallower layer that is *actively* mixing (Gray et al., 2020), the depth of the epilimnion indicates the volume and properties of the water that is influenced by air-water interactions. It is therefore essential for interpreting the physical response of lakes to long-term atmospheric changes (Lorbacher et al., 2006, Persson and Jones, 2008, Flaim et al., 2016), and extreme climatic events (Jennings et al., 2012, Calderó-Pascual et al., in review) and is even required for predicting the local climate for very large lakes (Thiery et al., 2015). The epilimnion depth is also critical for the estimation of algal light availability, nutrient fluxes and epilimnetic water temperatures, which determine photosynthesis rates and establishes the basis of the food web in a lake (MacIntyre, 1993, Diehl et al., 2002, Berger et al., 2006, Bouffard and Wüest, 2018). The depth of the epilimnion is also used for estimating the transfer of oxygen, received at the lake surface, to deeper layers, sustaining aerobic life and preventing anoxia (Foley et al., 2012, Schwefel et al., 2016).

The increasing availability of high-frequency measured and simulated data, coupled with collaborative networks of lake scientists, offers a huge potential for broadening our understanding of the epilimnion depth. Water temperature profile data collected at high frequency intervals on automatic monitoring buoys in lakes are becoming increasingly available (Jennings et al., 2012, de Eyto et al., 2016, Marcé et al., 2016). In addition, the collation of these datasets globally through collaborative initiatives such as GLEON (http://gleon.org/) and NETLAKE (https://www.dkit.ie/netlake) (Weathers et al., 2013, Jennings et al., 2017), and modelling initiatives such as ISIMIP2b (Ayala et al., in review), broadens the potential for long-term and multi-lake studies. However, these datasets also introduce new challenges for estimating metrics such as the epilimnion depth. Such large quantities of big data can limit users' capacity to examine individual profiles and therefore require robust, automated approaches with low computational expense (Read et al., 2011, Pujoni et al., 2019).

Despite the ubiquity of the epilimnion depth, there is no objective or consistent method used in limnology. The epilimnion depth can be defined in terms of many variables (e.g. water temperature, water density, turbulence estimations, surface fluxes, biogeochemical properties), represent different temporal scales of variability (e.g. inter-annual to sub-daily), and be calculated using a range of numerical approaches (e.g. sigmoidal functions, threshold approaches) (Brainard and Gregg, 1996, Thomson and Fine, 2003, Kara et al., 2003, De Boyer et al., 2004, Lorbacher et al., 2006, Gray et al., 2020). A particularly common approach in limnology, due to the availability of the required data, is to define the epilimnion using water temperature profile data. However, inconsistencies exist between studies which use water temperature (e.g. Zorzal-Almeida et al., 2017, Strock et al., 2017), or water density (e.g. Read et al., 2011, Obrador et al., 2014). Often the epilimnion depth is defined as the location where the change in water temperature or density exceeds a user-defined threshold. However, studies vary in the value selected which may be defined in absolute units (e.g. Andersen et al., 2017) or gradients between consecutive sensors (e.g. Lamont et al., 2004). A particularly prevalent method in recent studies is the 'meta.top' function proposed in R package 'rLakeAnalyzer' (Read et al., 2011). In contrast, vertical turbulence data are not routinely measured





in lakes, and therefore definitions based on actual turbulence measurements are uncommon. However, vertical turbulence profiles, as well as water temperature profiles, are estimated by some hydrodynamic lake models (Goudsmit et al., 2002, Dong et al., 2019). Such modelled data, therefore, offers a tool for assessing commonly used water temperature/density based methods in comparison to turbulence based methods.


The diversity of epilimnion depth definitions and arbitrary selection process, suggests that methods may be used interchangeably, and are relatively insensitive to the threshold value used. However, recent studies have begun to recognise large inconsistencies between different definitions and the potential problems this may cause, although so far, in limnology, analysis has been restricted to a small number of manual profiles (Gray et al., 2020) and a limited number of methods

(Pujoni et al., 2019). As such, a systematic analysis of common epilimnion depth methods for use with high-frequency data is required to assess the agreement among methods. Given the potential of multi-lake comparison and longitudinal studies, methods are required to perform consistently across temporal and spatial ranges, rather than being tailored specifically to one lake or period of time. Therefore, the sensitivity of different methods to temporal and spatial characteristics, such as water column structure and vertical resolution of data measurements is essential for assessing which methods are most suitable for

future analysis (Fee et al., 1996, Thomson and Fine, 2003,Lorbacher et al., 2006, Pujoni et al., 2019).

In this study, we undertook an in-depth comparison of methods commonly used for the estimation of epilimnion depth using high frequency, multi-year data for water temperature profiles, collected with automated monitoring buoys from two European lakes, Lough Feeagh (Ireland) and Lake Erken (Sweden). In addition to estimates based on these measured data,

we used simulated data output from a lake model to compare water temperature and turbulence based methods, and to assess the influence of vertical sensor resolution. The objectives of this study were to: 1) compare water temperature and water density based estimates of the epilimnion depth, 2) compare a range of common methods and threshold values, 3) assess the sensitivity of individual methods to the threshold value, the water column structure, and the vertical sensor resolution, and 4) to compare profile based methods to turbulence derived estimates using lake modelled data.

**2    Methods**

**2.1 Study sites**

We used data from two European temperate lakes, Lough Feeagh (53°56'N, 9°34'E) in Ireland and Lake Erken (59°51'N, 18°36'E) in Sweden (Fig.1.). Lough Feeagh is located on the west coast of Ireland and is a cold monomictic, oligotrophic and humic lake with a surface area of 3.9 km$^2$, maximum depth of 45 m and average depth of 14.5 m (de Eyto et al., 2016).

Lake Erken is located in east central Sweden near the Baltic coast and is a dimictic, mesotrophic, clear lake with a surface area of 24 km$^2$, maximum depth of 21 m and average depth of 9 m (Yang et al., 2016). The lakes differ in many characteristics, including depth, surface area and sensor deployment resolution, providing an opportunity to assess method performance in different lake specific conditions. In addition, Lake Erken has a much larger average summer top-bottom density gradient (0.056 kg m$^{-3}$ m$^{-1}$) compared to Lough Feeagh (0.016 kg m$^{-3}$ m$^{-1}$).

**2.2 Measured data**

In this study, we used a total of 4783 daily water temperature profiles from Lough Feeagh (n = 2778) and Lake Erken (n = 2005). Profiles were collected at high frequency intervals on moored automatic monitoring buoys, and from these the average daily profiles were calculated. On Lough Feeagh, vertical water temperature measurements were collected every 2 minutes for the period 2004-2017 at depths 0.9, 2.5, 5, 8, 11, 14, 16, 18, 20, 22, 27, 32, 42 m using submerged platinum

resistance thermometers (PRTs) (PT100 1/10DIN, Lab Facility, Bognor Regis, United Kingdom) (de Eyto et al., 2016,



2020). On Lake Erken, temperature profile data were collected at 1 min intervals at depths 0.5 m to 15 m at 0.5 m intervals, using Type T thermocouple sensors using a Campbell scientific AM416 multiplexer and CR10 datalogger (Pierson et al., 2011). The topmost sensor data was excluded to match the topmost sensor in Lough Feeagh. In Lake Erken, the monitoring buoy was manually deployed each year prior to or just after the onset of stratification to avoid damage from the seasonal ice

cover, and therefore the number of observations varied annually. To ensure data were consistent for both lakes, data were subset from 1st April to 31st October. To address the issue of large data gaps, years where less than 70% of the data between April to October were available (>150 days) were excluded from the analysis. The remaining years were 2004, 2005, 2006, 2011, 2012, 2013, 2014, 2015, 2016, 2017 for Lough Feeagh and 2002, 2005, 2008, 2009, 2014, 2015, 2017 for Lake Erken. Water density (kg m$^{-3}$) was calculated from water temperature (°C) using rLakeAnalyzer (Read et al., 2011), with the Martin

and McCutcheon (1999) equation, assuming negligible effects of soluble material.

Meteorological data were required to drive a physical hydrodynamic model (GOTM; Global Ocean Turbulence Mode, Burchard et al,. 1999), including wind speed (m s$^{-1}$), atmospheric pressure (hPa), air temperature (°C), relative humidity (%), cloud cover (dimensionless, 0-1), short-wave radiation (W m$^{-2}$) and precipitation (mm day$^{-1}$). For Lake Erken, air

temperature, wind speed and short-wave radiation were collected from the Malma Island meteorological station on the lake, at 1 min intervals and averaged to 60 min intervals. Mean sea level pressure, relative humidity and precipitation were measured at the Svanberga meteorological station located 400 m from the lake shore, at 60 min intervals. Cloud cover was recorded from Svenska Högarna Station, 69 km south-east of Lake Erken. In Lough Feeagh, wind speed, air temperature and short-wave radiation, mean sea level pressure, relative humidity and precipitation were measured in the meteorological

station next to the lake (de Eyto et al., 2020). Cloud cover was recorded at Knock Airport, 50 km east from Lough Feeagh.

### 2.3 Simulated data

The Global Ocean Turbulence Model (GOTM), adapted for use in lakes, simulates small-scale turbulence and vertical mixing (Burchard et al,. 1999, Sachse et al., 2014, Moras et al., 2019, Ayala et al., in review). GOTM was used to simulate daily profiles of water temperature (°C) and vertical eddy diffusivity (m$^{-2}$ s$^{-1}$) for 1016 days in Lough Feeagh (2012-2016)

and 1449 days in Lake Erken (2010-2016). For these simulations, the turbulent kinetic energy-dissipation (k-ε) model was used, in combination with the algebraic second-moment model (Canuto et al., 2001). The ACPy (Auto Calibration Python) program was used to calibrate the model and three non-dimensional scaling factors that affect the surface heat-flux, short-wave radiation input and wind were calibrated, as well as parameters affecting the minimum turbulent kinetic energy and e-folding depth for visible fraction of light. Overall, there was a good model fit for both lakes (Table 1).

### 2.4 Definitions for the epilimnion depth

We selected four epilimnion depth definitions that are commonly used in limnology and that were computationally efficient for multi-year automated high frequency data. These methods we describe as profile based methods (M1 – M4) (Fig.2.). In addition, we calculated epilimnion depth using a method for modelled data only (M5). In our analysis, epilimnion depth was expressed relative to the water surface and is therefore always a negative value. The range of thresholds used for each

method was selected based on the values found within the literature (see Table 1 in Gray et al., 2020). We made no assumption of the conditions below the deepest measured depth and therefore the deepest estimated epilimnion depth was limited to the maximum measured depth for each lake (42 m in Lough Feeagh and 15 m in Lake Erken).

*2.4.1 Absolute difference from the surface method (M1)*

In M1, the epilimnion depth was defined as the shallowest depth where the density was a given 'threshold' value more than

the surface density (Fig. 2.), with the surface density ($\rho_1$) approximated as the density at the topmost sensor deployment, 0.9 m in Lough Feeagh and at 1 m in Lake Erken. We used a linear interpolation method to estimate the epilimnion depth on a


continuous depth scale for all methods (Read et al., 2011), which assumed a linear relationship of densities between the first measured depth which exceeded the threshold ($z_{i+1}$) and the preceding measured depth ($z_i$). The numerical scheme can be described as (using notation from Read et al., 2011);

$$z_e = z_i + ((\rho_1 + \Delta\rho) - \rho_i)(\frac{z_{i+1} - z_i}{\rho_{i+1} - \rho_i}), \tag{1}$$

where $z$ is depth (m), $\rho$ is water density (kg m$^{-3}$), and $\Delta\rho$ is the threshold value (kg m$^{-3}$). The threshold values for the absolute method, M1 only, ranged from 0.025 kg m$^{-3}$ to 0.2 kg m$^{-3}$ at intervals of 0.025 kg m$^{-3}$. For all methods excluding the rLakeAnalyzer method (M4), if the threshold value was not exceeded, the epilimnion depth was defaulted to the deepest value (Lorbacher et al., 2006). Epilimnion depth estimates calculated with water temperature used the same type of equation (Eq. 1) but with temperature rather than density and noting that temperature decreases with depth. The only threshold value

used for temperature was 1 °C.

### 2.4.2 Gradient from the surface method (M2)

In M2, the epilimnion depth was defined as the shallowest depth where the density gradient between consecutive measured depths exceeded the threshold value. M2 can be described as,

$$z_e = z_{i\Delta} + \left(\Delta\rho/\Delta z - \frac{\partial\rho}{\partial z_{i\Delta}}\right)\left(\frac{z_{i\Delta+1} - z_{i\Delta}}{\frac{\partial\rho}{\partial z_{i\Delta+1}} - \frac{\partial\rho}{\partial z_{i\Delta}}}\right), \tag{2}$$

where $z_{i\Delta}$ is the midpoint between $z_i$ and $z_{i+1}$, and $\frac{\partial\rho}{\partial z_{i\Delta}}$ is the density gradient between $z_i$ and $z_{i+1}$ and $\Delta\rho/\Delta z$ is the threshold

value (kg m$^{-3}$ m$^{-1}$). The threshold values for all gradient methods, (i.e. M2, M3 and M4), ranged from 0.025 kg m$^{-3}$ m$^{-1}$ to 0.2 kg m$^{-3}$ m$^{-1}$ at intervals of 0.025 kg m$^{-3}$ m$^{-1}$.

### 2.4.3 Gradient from the pycnocline method (M3)

In M3, the epilimnion depth was defined as the deepest depth where the density between consecutive measured depths exceeded the threshold value, starting from the depth of the maximum density gradient (hereafter the 'pycnocline') as the

reference depth, and moving to successively shallower measured depths. M3 can be described by,

$$z_e = z_{i\Delta} + \left(\Delta\rho/\Delta z - \frac{\partial\rho}{\partial z_{i\Delta}}\right)\left(\frac{z_{i\Delta} - z_{i\Delta+1}}{\frac{\partial\rho}{\partial z_{i\Delta}} - \frac{\partial\rho}{\partial z_{i\Delta+1}}}\right). \tag{3}$$

### 2.4.3 rLakeAnlayzer (M4)

In M4, the epilimnion depth was defined using the rLakeAnalyzer function 'meta.depths' (relating to output "meta.top"), which used the same numerical scheme as M3, Eq. (3), but differed in certain assumptions (Read et al., 2011). Firstly, in M4, the epilimnion depth was prohibited from extending below the depth of the pycnocline. Therefore, for profiles where the

predefined threshold value was less than the maximum density gradient, the epilimnion depth defaulted to the maximum density gradient. This differed from the other methods where, for such profiles, the epilimnion depth was defaulted to the deepest measured depth. Secondly, a user defined filter ('mixed.cutoff' object) was used to remove profiles which were not sufficiently stratified to identify an epilimnion depth. We used the default filter value, which removed profiles where the overall water temperature range was less than 1°C. For the days which did not meet the filter value, and no epilimnion depth

was identified, we set the epilimnion depth to the deepest measured depth (i.e. no epilimnion depth) to ensure each method had the same number of data points for comparison with other methods.

### 2.4.5 Modelled turbulence method (M5)

The modelled turbulence method (M5) used the GOTM lake model simulated profile estimates of vertical eddy diffusivity (m$^2$ s$^{-1}$). In M5, the epilimnion depth was defined as the first depth where the vertical eddy diffusivity fell below the

predefined threshold value, and was described as;




$$z_e = z_i + \left(\Delta K_z - K_{z_i}\right)\left(\frac{z_{i+1}-z_i}{K_{z_{i+1}}-K_{z_i}}\right), \qquad (4)$$

where $K_z$ is vertical eddy diffusivity (m$^2$ s$^{-1}$) and $\Delta K_z$ is the threshold value (m$^2$ s$^{-1}$). The thresholds ranged from $10^{-5}$ to $10^{-4}$ m$^2$ s$^{-1}$ at intervals of $10^{-5}$ m$^2$ s$^{-1}$, based on the values described in Wüest and Lorke (2009) and MacIntyre and Melack (2009).

### 2.5 Analysis methods

To compare water temperature and water density based estimates of the epilimnion depth, we used M1 only and used a water temperature threshold value of 1 °C with a density threshold of 0.1 kg m$^{-3}$ for both sites. Firstly, we investigated the relationship between 1 °C and 0.1 kg m$^{-3}$ throughout the year. To do this, we calculated the long-term average water column temperature for each Julian day. For each day, we then calculated the change in density that would result from a 1 °C increase in the water temperature. We then subtracted 0.1 kg m$^{-3}$ from each Julian day value. Positive values, shown in red, indicated that a 1 °C increase in the water temperature, resulted in a greater than 0.1 kg m$^{-3}$ change in water density, while negative values, shown in blue, indicated a less than 0.1 kg m$^{-3}$ change in water density. Secondly, we compared water temperature and water density based estimates of the epilimnion depth. To do this we calculated the difference between the average water density derived estimate and the water temperature derived estimate for each Julian day. Positive differences, shown in red, indicated that the water density derived estimate was shallower than the water temperature derived estimate, while negative values, shown in blue, were deeper. For all analysis of measured data, the total number of observations were used for Lough Feeagh (n= 2778, years = 10) and Lake Erken (n = 2005, years = 7).

Following this, we confined the analysis to comparing water density based epilimnion depth estimates, using all four methods: M1, M2, M3 and M4 and the range of thresholds described earlier. Using data from both sites, we considered overall variability (i.e. how much do estimates vary between all methods and all thresholds?), variability within each individual method using different threshold definitions (i.e. how sensitive are estimates to the threshold value selected?) and variability between methods (i.e. what systematic differences exist between pairs of methods?). Given that we had a total of 32 time series to compare, 4 methods each with 8 threshold values, it was necessary to compute summary statistics for each of them. Therefore, the following statistics were calculated for all 32 time series, for the period from 1$^{st}$ April to 31$^{st}$ October each year and then averaged across all years. Firstly, we calculated the average epilimnion depth and presented the values for all methods and all thresholds values. We also summarised these statistics for each method, showing the average, minimum (shallowest), maximum (deepest) and range for each method, to demonstrate differences between methods. A large range in epilimnion depth estimates, indicated high sensitivity to the threshold value. Secondly, we calculated the percentage of days with available data, where the epilimnion depth was detected above the deepest measured depth. This demonstrated differences between methods in regards to the stratified period. Thirdly, we calculated the percentage of days with available data where the epilimnion depth was detected above the maximum density gradient or pycnocline. By definition the epilimnion should have relatively small density gradients and should not be equal or deeper than the pycnocline, however automated methods, depending on the logical and numerical schemes used to calculate them, may encroach on the metalimnion (Lorbacher et al., 2006). We therefore used this metric to investigate how frequently epilimnion depth estimates calculated by each method erroneously extended into the metalimnion.

Pearson's correlation coefficients were also calculated for all possible combinations between the 32 time series, to quantify the degree of association between them, without using any estimates of significance (Thomson and Fine, 2003, Rivetti et al., 2016). We presented only the average Pearson's correlation coefficient for each method, representing the average correlation for all possible combinations between threshold values. This indicated the extent to which changing the threshold value





influenced the temporal patterns. We also presented the average Pearson's correlation coefficients between each pair of methods (e.g. for all threshold combinations between M1 and M2 etc.) to demonstrate method agreement.

We also assessed the sensitivity of the profile based methods to changes in the water column structure and the vertical sensor resolution of measured data. For the water column structure sensitivity analysis, we calculated the long-term average

epilimnion depth estimate for each Julian day for all 32 method/threshold time series. For each method, using all thresholds, we calculated the range for each Julian day. The range in estimates was presented alongside the top-bottom density gradient for each Julian day, to investigate whether threshold sensitivity varied temporally and with water column structure. For the vertical sensor deployment resolution sensitivity analysis, we compared simulated water density profiles for both lakes at two different resolutions. High resolution data was resolved to 0.5 m for both lakes. Low resolution data were subset to an

average of 1 sensor per 3 m, using the measured depths for Lough Feeagh, and data from 1, 2.5, 5, 8 and 13 m for Lake Erken. We then calculated the difference between the Apr-Oct average epilimnion depth for the high and low resolution data. Methods where the high and low resolution data produced very different estimates were regarded as having high sensitivity to the vertical resolution of the data, while methods with small differences indicated low sensitivity. For all analysis using simulated data, the total number of observations were used for Lough Feeagh (n = 1016, years = 5) and Lake Erken (n =

1449, years = 7).

Finally, we assessed how each profile based method compared against the turbulence based estimates. For this analysis, both water density and vertical eddy diffusivity profile data were derived using the GOTM lake model. Then, using the same procedures as the measured data, we calculated the Apr-Oct average epilimnion depth for each method. We then calculated

the difference between the turbulence method (M5) and each of the four profile based methods (M1- M4). We also presented the average Pearson's correlation coefficients between each method and M5 (e.g. for all threshold combinations between M5 and M1, etc.). These results indicated the extent to which profile based methods were able to characterise active mixing penetration, within a hydrodynamic model setting, rather than confirming which method was more reliable for predicting the 'true' mixing depth.

**3    Results**

**3.1 Comparison between a water temperature and water density derived method**
There were large systematic differences between the epilimnion depth calculated using a water temperature based method compared to values calculated using a water density based method (Fig. 3). Due to the non-linear relationship between water density and water temperature, the difference in density induced by a water temperature increase of 1°C (water column

average) varied seasonally, with the pattern differing between sites (Fig. 3a). We found that on average, during the spring (April-May), when water column temperatures in both lakes were relatively low, a change of 1°C resulted in a water density change of less than 0.1 kg m$^{-3}$. As a result of this anomaly, estimates of the epilimnion depth that were based on water temperature data were shallower compared to those calculated using the water density method (Fig.3b). In contrast, in general from June to October for both sites, a change of 1°C in water temperature induced a change in water density of

greater than 0.1 kg m$^{-3}$, which resulted in estimates of the epilimnion depth which were deeper when using water temperature compared to those estimated using water density. Based on the long-term daily averages, the differences in the estimates of epilimnion depth between the two methods ranged from 3 to 5 m for Lough Feeagh, and 2 to 4 m for Lake Erken.



### 3.2 Comparison between water density based methods (M1 – M4)

Inspection of water column profiles highlighted key differences in the performance of methods M1, M2, M3 and M4 (Fig.
4). In a stratified profile, with a well-defined three-layered water column profile, there was often strong agreement on the
epilimnion depth between all methods and thresholds (Fig. 4a). In contrast, when the measured temperature profile was more
complex, i.e. at times when there was some stratification close to the surface or when a secondary pycnocline had developed
close to the surface, there was less agreement on the estimates of epilimnion depth between methods (Fig. 4b). For such
profiles, estimates of the epilimnion depth calculated with the absolute difference from the surface method, M1, were
typically staggered at linear intervals along the profile depending on the exact threshold value. In contrast, estimated
epilimnion depth calculated using the gradient methods (M2, M3 and M4) had a tendency to cluster at discrete locations on
the profile. Therefore, a small change in the threshold value induced either no difference at all in the epilimnion depth or at
other times a very large difference. For profiles with low water column stability, there was particularly large differences in
the estimated epilimnion depth calculated using different methods, reflecting differing underlying assumptions (Fig.4c). For
M3, for example, the epilimnion depth was defaulted to the deepest depth when the threshold value was not exceeded, as
was also the case for methods M1 and M2. In contrast, however, in M4, near-isothermal profiles often met the 'mixed.cutoff'
filter condition (i.e. water column range > 1°C), whilst still not having sufficient density gradients to meet the user threshold
value. As a result, in M4, the epilimnion depth was defaulted to the pycnocline, which, given the small density gradients,
was often found at a very shallow depth.


Time series results demonstrated the extent of the variability in epilimnion depth estimates between all methods and
thresholds (Fig. 5). Considering that all the time series estimates for Lough Feeagh (left-side) and Lake Erken (right-side)
were presumed to estimate the same theoretical location, they would ideally all produce exactly the same temporal patterns.
Instead, the differences were large enough to obscure the annual patterns and hinder the ability to compare between the two
lakes. The overall variability between all estimates was particularly high for Lough Feeagh, where the Apr-Oct average
epilimnion depth ranged by 36.9 m (-4.6 m to -41.5 m) while in Lake Erken, estimates ranged by 5.2 m (from -7.8 m to -13.0
m) (Fig.6a).

There were evident systematic differences between methods. In both lakes, the average Apr-Oct epilimnion depth for each
method was shallowest for M1 and was on average shallower by 17.0, 16.6 and 2.2 m compared with methods M2, M3 and
M4 in Lough Feeagh, and 1.2, 1.7, 0.8 m in Lake Erken (Table 2.). The minimum (shallowest) estimates of the Apr-Oct
average, for gradient methods (M2, M3 and M4) were comparable in magnitude to the maximum (deepest) estimate for the
absolute difference from the surface method, M1. The average Pearson's correlation coefficient between each pair of
methods also demonstrated that certain method pairs had greater temporal agreement than other pairs (Table 3). Method
pairs, M3-M2 and M4-M1 had particularly high Pearson correlation coefficients for both lakes, suggesting these methods
produced similar temporal trends. In Lake Erken all method pairs had higher Pearson's correlation coefficients than Lough
Feeagh.

The selection of a threshold value proved to be very important in the estimation of the epilimnion depth. For all methods,
smaller threshold values produced shallower estimates of the average Apr-Oct epilimnion depth while larger threshold
values produced deeper estimates (Fig.6a). Methods with a large range between the shallowest (minimum) and deepest
(maximum) estimate demonstrated high sensitivity to the threshold value (Table 2). For both lakes, the range in the average
Apr-Oct epilimnion depth estimates for each method was very high for M2, M1 and M3, indicating high threshold sensitivity
in these methods. Method M4 had a substantially lower range than all other methods and a very high average Pearson's
correlation coefficient, indicating that both the average value and the temporal pattern of the epilimnion depth were only





weakly influenced by the threshold value. In both lakes, methods M2 and M1, where the epilimnion depth was defined from the surface downwards, had a higher range in estimates calculated with different threshold values, compared to methods M3 and M4, where the epilimnion was defined from the pycnocline upwards. M1, however, had higher average Pearson's correlation coefficient than M2 and M3, indicating that the temporal pattern of the epilimnion depth was less influenced by the threshold value. In general, the threshold sensitivity of each method reduced with increasing threshold size. That is, the changes in the epilimnion depth occurring between threshold values decreased with increasing threshold value (Fig.6a). For example, for M2, the difference in the Apr-Oct average epilimnion depth between the first two thresholds (0.025 and 0.05 kg $m^{-3}$ $m^{-1}$) was much greater than the difference between the last two thresholds (0.175 and 0.2 kg $m^{-3}$ $m^{-1}$), in both lakes.

The percentage of stratified days, defined as days where the epilimnion depth was identified at a depth greater than the deepest measured depth, demonstrated the extent to which different methods/thresholds influenced annual patterns (Fig.6b). For M4, the percentage of stratified days remained static regardless of the threshold value for method M4 as an epilimnion was defined whenever the water column temperature range was more than 1 °C regardless of the threshold used. For all other methods, the number of stratified days decreased with increasing threshold value. For M1 the difference in stratified days between threshold values was small, compared to both gradient methods M2 and M3, particularly in Lough Feeagh. For example, in Lough Feeagh, for M3, the number of stratified days calculated using a threshold value of 0.25 kg $m^{-3}$ $m^{-1}$ was 125, while for threshold values greater than 0.075 kg $m^{-3}$ $m^{-1}$, the average number of stratified days per annum decreased to less than 38 days.

The percentage of days where the epilimnion depth was located above the pycnocline, defined as days where the epilimnion depth was identified above the maximum density gradient, indicated that some methods may be less prone to erroneously estimating the epilimnion depth in the metalimnion, compared with others (Fig.6b). For both lakes, M1 had the highest number of days where the epilimnion depth was located above the pycnocline, suggesting that on average the method extended into the metalimnion less frequently than other methods. In Lough Feeagh, all gradient methods, M2, M3 and M4, had very high range occurring between the different threshold values. In Lough Feeagh, gradient methods calculated with a threshold value greater than 0.15 kg $m^{-3}$ $m^{-1}$, resulted in an average of zero days where the epilimnion depth was located above the pycnocline.

### 3.3 Sensitivity of epilimnion depth to water column structure

For all methods, threshold sensitivity fluctuated seasonally, although varied in pattern (Fig. 7). Threshold sensitivity was shown by the range between the shallowest and deepest epilimnion depth estimates calculated for all threshold values. In Lough Feeagh, M1 had a smaller range in epilimnion depth estimates during the peak summer months of June, July and August, compared with months when the onset and overturn of stratification commonly occurred. During periods of transient stratification, the stability of the water column was often low but frequent changes in the near-surface water density, induced large differences between estimates calculated using small thresholds compared with large threshold values. In contrast, methods M2 and M3 had the highest range in estimates occurring during the peak summer months. Even during peak summer in Lough Feeagh, gradients in the water column were relatively small (Fig. 7b), which resulted in a very large range between the smallest threshold values which found a near-surface epilimnion depth, and the largest thresholds that often found no epilimnion depth at all, therefore defaulting to the deepest depth. In Lake Erken, the water density gradients were typically much larger, and methods M1, M2 and M3 all peaked during May and June, when gradients in the water column were typically increasing but prone to fluctuations. For both lakes, M2 had typically a higher threshold range than M3 during peak summer and the overturn period, which was related to the common development of a secondary pycnocline. M4 produced much lower ranges in the epilimnion depth throughout the year, since as long as the 'mixed.cutoff' filter was met, the epilimnion depth was defaulted to the pycnocline if the threshold was not exceeded, thus largely reducing the ability for





large differences to occur. The range in epilimnion depth estimates for M4 was highest during the peak summer months,
which was when the epilimnion depth was typically shallowest and more frequently defined by the threshold value rather
than defaulting to the pycnocline.

### 3.4 Sensitivity of epilimnion depth to vertical sensor resolution

The vertical resolution of water density data was found to have a systematic influence on the estimation of the epilimnion
depth for all methods (Table 4). Overall, the modelled higher vertical resolution data resulted in shallower estimates of the
epilimnion depth, relative to the estimates made with the modelled low resolution data. For Lough Feeagh, the results
showed that the annual average Apr-Oct epilimnion depth estimate using high resolution data were on average 0.1, 3.2, 3.2
and 0.5 m shallower than those using low resolution data for methods M1, M2, M3 and M4 respectively, while in Lake
Erken they were 0.0, 1.2, 1.0, 0.2 m shallower. Methods M1 and M4 had substantially smaller differences between high and
low resolution estimates compared with M2 and M3. In particular, M1 had almost no difference between high and low
resolution data, indicating that this method had very low sensitivity to the vertical sensor deployment.

### 3.5 Comparison with modelled turbulence method (M5)

In general, the modelled turbulence method had very low sensitivity to the threshold value, compared with the profile based
methods also calculated using modelled data. For both lakes, we found that the modelled turbulence method produced
shallower estimates than modelled profile based methods (Table 5). In Lough Feeagh, the average Apr-Oct epilimnion depth
estimate using the modelled turbulence method M5 was -20.8 m, which was 1.3 m, 11.0 m, 11.2 m and 1.3 m shallower than
methods M1, M2, M3, and M4 respectively, while in Lake Erken the M5 estimate was -11.0 m, which was 0.0 m, 1.0 m, 1.1
m and 0.4 m shallower. In both lakes, M1 had the strongest agreement with M5, demonstrated by both the average difference
(1.3 m in Lough Feeagh and 0.0 m in Lake Erken), and the highest Pearson's correlation coefficient in Lough Feeagh (r =
0.90) and Lake Erken (r = 0.89). This was followed by M4, which also had strong agreement with M5. In contrast, M2 and
M3 had much weaker agreement with M5, in terms of both the Apr-Oct epilimnion depth estimate and the Pearson's
correlation coefficients.

### 4    Discussion

The concept of the epilimnion, and more widely, the three-layered structure of a stratified lake, is fundamental in limnology.
Yet, despite the ubiquity of the term, there is no objective or generic approach for defining the epilimnion and a diverse
number of approaches prevail in the literature. In a comprehensive analysis of high-frequency, multi-year data from two
lakes, this study has highlighted the extent to which common water temperature profile based epilimnion depth estimates
differ. The level of variability in epilimnion depth estimates calculated using common methods and threshold values, was
exceedingly high. This result calls into question the practice of arbitrary method selection and comparing findings between
studies which use different methods or even just different thresholds. The magnitude of variability also casts ambiguity on
the calculation of key biogeochemical and ecological processes in a lake that rest on the assumption that the layers of a lake
are well defined, including calculations of metabolic rates, and oxygen fluxes (e.g. Coloso et al., 2008, Foley et al., 2012,
Obrador et al., 2014, Winslow et al., 2016). Ultimately, these results emphasise the limitations inherent to defining the
epilimnion using water temperature profile data and on a temporally continuous basis. In an idealised stratified profile, the
epilimnion is portrayed as near-uniform in water temperature or density and clearly delineated from a well-defined
metalimnion. However, the vast majority of the measured profiles, at least within this study, did not conform to this idealised
three-layered structure. In these cases, methods not only diverged on the location of the epilimnion depth but also may not
even be underpinned by the same theoretical principles. Since none of these methods can be considered the 'true' definition





of the epilimnion depth, acknowledgement of the systematic differences between methods, manifest over high-frequency time series, is essential for selecting suitable methods for future analysis.

A particularly distinct systematic difference was found between water temperature and water density. Due to the non-linear relationship between water density and temperature, the use of water temperature was equivalent to using different threshold values throughout the year, resulting in a distinct shift in the stratification period. Although water density gradients are driven by temperature changes in lakes and are also calculated from water temperature estimates, water density directly influences mixing processes and is therefore recommended for estimating the epilimnion depth (Read et al., 2011, Gray et

al., 2020). The implications of using a water temperature based method may be particularly enhanced in Northern temperate lakes due to the large annual water temperature ranges (Maberly et al., 2020). Pronounced differences in the estimation of the epilimnion depth were also found within estimates derived using the same water density input data. Typically, for the range of common thresholds used in this study, the absolute difference from the surface method, M1, produced shallower estimates relative to gradient based methods. In addition, the difference between these methods was particularly large when

the vertical resolution of the data was low. This suggests that studies using gradient based methods, particularly those using coarse vertical data, may have a deep bias relative to those using an absolute method, and consequently, were more prone to erroneously extending into the metalimnion. In addition, as may be expected, the use of larger threshold values also produced systematically deeper estimates of the epilimnion depth. Surprisingly, however, the magnitudes of these differences were on par with those occurring between methods. The implications of a shallow or deep bias may be far-

reaching, particularly given that various biological and ecological metrics have already been found to be highly sensitive to changes in the epilimnion depth (Coloso et al., 2008, Gray et al., 2020). For example, a deeper estimate of the epilimnion depth would systematically lead to a larger ratio between the epilimnion and euphotic depth, compared with a shallower estimate, which if used to understand the development of a phytoplankton bloom, could lead to contradictory results (Huisman et al., 1999). Alternatively, an acceleration of epilimnion shallowing during stratification onset, as found with

water temperature estimates, might indicate a longer duration of conditions necessary for phytoplankton growth The implications of a seasonal or deep/shallow biases may be even more important for computing fluxes (e.g. oxygen or nutrients) between the epilimnion and the metalimnion, since both terms are influenced by the epilimnion depth (Giling et al., 2017, Gray et al., 2020).

An important difference was also found between methods detecting the layer that is isothermal relative to the surface and

methods detecting the top of the metalimnion, which has not been well considered in the literature. M1 and M2, defined from the surface downwards, were more prone to the detection of a shallow secondary pycnocline, compared with M3 and M4. Instead, M3 and M4, defined from the pycnocline upwards, prioritised the relative difference between the metalimnion and the surface. From a theoretical point of view, processes related to the air–water interface could be better suited to methods identifying the isothermal layer, while for processes related to the entrainment of deep water into the epilimnion are

more suited to top of the metalimnion methods.

The selection of an epilimnion method also had surprisingly large consequences for understanding the stratification period, which is widely used for quantifying the impact of climate change on lakes (Livingstone, 2003, Butcher et al., 2015, Ayala et al., in review). Notably, the average epilimnion depth and number of stratified days calculated using M4, depended very little on the threshold value selected. Instead, the selection of the filter (defaulted to a water column range of > 1°C), which

was unique to this method, determined the number of stratified days and largely influenced the other bulk statistics. This also resulted in the epilimnion being identified even when the threshold was not exceeded, which in some instances could have the effect of muting relative temporal changes in the epilimnion depth. In contrast, for the other methods, the threshold was used to determine whether the water column was considered to be stratified and therefore the stratification period was highly



sensitive to the threshold value, similarly to the other bulk statistics. Ultimately, these results suggest that the stratification
period calculated in different studies or for different regions cannot be compared unless identical definitions are used. The
method most appropriate for identifying the stratified period has been considered in other studies (Woolway et al., 2014,
Engelhardt and Kirillin, 2014) however our results offer some additional insights. Although water temperature thresholds are
typical for defining the stratification period in lakes, they may not be suitable for use with water density metric and
potentially introduce a seasonal bias. In addition, estimations of the epilimnion depth, and the variability among definitions,
may be particularly relevant for understanding the stratified period since it is often assumed that the onset of stratification
marks the decoupling of the epilimnion from the deeper layers, thus determining the duration of nutrient limitations in the
epilimnion and oxygen limitations in the hypolimnion (MacIntyre, 1993, Foley et al., 2012, Schwefel et al., 2016).

Regardless of the method selected, however, all water temperature/density based methods are limited in their ability to
indicate actual mixing processes. Our results using the lake modelled turbulence data demonstrated that even in a modelled
environment, methods were inconsistent with turbulence based methods, which generally resulted in a shallower epilimnion
depth estimate. Although these results are not necessarily indicative of measured data, they do highlight the need for caution
when interpreting water temperature/density derived epilimnion depth estimates. There is an important but subtle difference
between the layer that has been recently well-mixed and therefore has little resistance to further mixing, due to the lack of
density gradients, and the layer that is *actively mixing* and is determined only through directly measured turbulence. Many of
the ecological applications of the epilimnion depth have the underlying assumption that enough mixing is occurring in the
epilimnion to keep the relevant organisms or particles suspended within the layer. However, whether mixing is actually
occurring and to what extent, is not described by epilimnion depth estimations derived using water temperature or density
profile data, and in fact, previous studies have found water density estimates of the epilimnion depth to be relatively poor
indicators for the homogeneity of other ecological variables (Gray et al., 2020). Therefore, epilimnion depth estimates
derived from profile data may be more reliably used for indicating relative changes in mixing over time (Read et al., 2012,
Calderó-Pascual et al., in review).

The selection of a suitable threshold value is far more important than previously attributed in limnology. In general, a
suitable threshold is any value that can be reasonably considered as homogenous while also within the limit of sensor
detection (De Boyer et al., 2004). However, all the threshold values used in this study met these criteria, yet produced
fundamentally different epilimnion depth estimations and temporal patterns. Although, it may be unreasonable to suggest a
'universal' threshold value, a given study may find a threshold that is less problematic than other values. In general, we
found that the sensitivity of the epilimnion depth to the threshold value decreased with the increasing size of the threshold.
That is, for small thresholds the impact of changing the threshold value was greater than larger thresholds for the same
incremental change. This may suggest that for studies using smaller threshold values, the results are more threshold
dependent than those using large threshold values. However, larger threshold values had greater frequency of the epilimnion
depth estimates being below the maximum density gradient, suggesting that larger threshold values tend to extend into the
stable depths of the metalimnion more regularly, and hence somewhat explaining the lower threshold sensitivity. The trade-
off between threshold sensitivity and encroachment into the metalimnion points towards a mid-range threshold, such as 0.1
kg m$^{-3}$ or 0.1 kg m$^{-3}$ m$^{-1}$, as potentially, being more reliable than large or small thresholds.

One of the main goals behind the global collection of high-frequency data in lakes is to understand how physical processes
vary between lakes, which indicate how different lakes may respond to changing climatic conditions (Weathers et al., 2013,
Kraemer et al., 2015, Woolway et al., 2019). In order to understand this, we require methods that perform consistently
between lakes and over longitudinal scales. The differences between the two lakes studied, in particular variability in water
column structure, the strength of density gradients and the vertical resolution of sensor deployment, influenced the level of





agreement between epilimnion depth methods. Overall, Lake Erken had much greater agreement among methods than Lough Feeagh. In particular, we found this to be related to the difference in vertical resolution of the measured data between sites. Of all the methods considered in this study, our results suggest that the absolute difference from the surface method, M1, might be more useful as a 'generic' method, due in particular to the very low sensitivity to the vertical sensor resolution compared with all other gradient based methods. This finding is in agreement with previous oceanography studies that have similarly found gradient methods to be highly sensitive to vertical resolution (e.g. Lorbacher et al., 2006, Thomson and Fine, 2003). In addition, however, the performance and threshold sensitivity of all methods also fluctuated temporally as influenced by changes in the water column structure. Assessment of the uncertainty associated with epilimnion depth estimates may be useful, particularly for studies comparing the epilimnion depth between periods of time that vary in stratification intensity.

Although long-term epilimnion depth trends are only rarely reported directly (e.g. Hondzo and Stefan, 1993, Fee et al., 1996, Sahoo et al., 2013), they are embedded in our understanding of many climate related variables. For example, the epilimnion depth plays a key role in modulating the effects of eutrophication, browning and climate change on lake water surface and epilimnetic temperatures (Persson and Jones, 2008, Flaim et al., 2016, Strock et al., 2017, Bartosiewicz et al., 2019). As such, changes in the epilimnion depth may enhance or mute the effect of increasing incoming heat on water surface temperatures and therefore may be particularly important in explaining temporal and spatial anomalies in surface temperature trends. Given the results of this study it may be that long-term trends calculated using different metrics relate to fundamentally different parts of the water column that may be undergoing different changes due to climate change. Therefore, the strength, and even the direction, of long-term trends in the epilimnion may be highly dependent on the definition used (Yang and Wang, 2008, Somavilla et al., 2017).

## 5 Conclusion

This study has demonstrated the extent to which different definitions of the epilimnion depth lead to different locations of the epilimnion depth in the water column and produce very different and contradictory temporal patterns. These results have wide-reaching relevance in limnology, including for studies interested in metabolism, eutrophication, and hypolimnetic anoxia. The sensitivity of epilimnion depth methods to temporal and spatial characteristics, such as morphology, water column structure and vertical resolution of data measurements may also pose challenges for studies interested in long-term trends or global lake comparison studies. While there is no prescribed rationale for selecting a particular method, the M1 method, defined as the shallowest depth where the density was 0.1 kg m$^{-3}$ more than the surface density, was shown to be overall less problematic than the other methods, and may be useful as a generic method.

## Code and data availability

The analysis codes and output data are stored in HydroShare http://www.hydroshare.org/resource/26dbc260405b4bb9b3ac16ec55432684. Source code of the model GOTM is freely available online at https://gotm.net/. The data used in this study from Lough Feeagh is available online at https://doi.org/10.20393/6C4760C2-7392-4347-8555-28BA0DAD0297.



**Competing interests**

The authors declare that they have no conflict of interest.

**Acknowledgements**

This research was carried out as part of the MANTEL Innovative Training Network which has received funding from the
European Union's Horizon 2020 research and innovation programme under the Marie Skłodowska-Curie grant agreement
No 722518. The long term monitoring program of Lough Feeagh is enabled by the field staff of the Marine Institute, and we
gratefully acknowledge their support. The monitoring program at Lake Erken received financial support from the Swedish
Infrastructure for Ecosystem Science (SITES)

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





**Tables**

**Table 1.** Lake model performance evaluation, showing the percentage relative error (%), root mean squared error (°C), and Nash Sutcliffe efficiency, for Lough Feeagh (profiles = 1016, years = 5) and Lake Erken (profiles = 1449, years = 7).


| Statistic | Lough Feeagh Calibration | Validation | Lake Erken Calibration | Validation |
|---|---|---|---|---|
| PRE (%) | -0.48 | 0.47 | -1.85 | 1.36 |
| RMSE (°C) | 0.67 | 1.18 | 0.53 | 0.55 |
| NSE | 0.97 | 0.92 | 0.98 | 0.97 |


**Table 2.** Summary of statistics for each method, showing the average (m), minimum (i.e. shallowest estimate) (m), maximum (i.e. deepest estimate) (m) and range (m) of the April-October epilimnion depth estimates (summarised from the results shown in Fig.6a), and the average Pearson's correlation coefficient (r) for each method, representing the average correlation for all possible combinations between threshold values., for Lough Feeagh and Lake Erken.

| Method | Lough Feeagh Average (m) | Min (m) | Max (m) | Range (m) | r | Lake Eken Average (m) | Min (m) | Max (m) | Range (m) | r |
|---|---|---|---|---|---|---|---|---|---|---|
| M1 | -18.9 | -4.6 | -25.4 | 20.8 | 0.77 | -10.0 | -7.8 | -11.2 | 3.4 | 0.92 |
| M2 | -35.9 | -19.7 | -41.4 | 21.7 | 0.48 | -11.3 | -8.4 | -12.9 | 4.5 | 0.78 |
| M3 | -36.5 | -22.4 | -41.5 | 19.1 | 0.49 | -11.8 | -10.0 | -13.0 | 3.0 | 0.82 |
| M4 | -21.1 | -19.7 | -21.5 | 1.8 | 1.00 | -11.9 | -10.1 | -11.3 | 1.2 | 0.99 |


**Table 3.** Average of all Pearson's correlation coefficients calculated for each pair of methods (e.g. for all threshold combinations between M1 and M2 etc.), for Lough Feeagh and Lake Erken.

| Method | Lough Feeagh r | Lake Erken r |
|---|---|---|
| M2-M1 | 0.35 | 0.77 |
| M3-M1 | 0.33 | 0.74 |
| M3-M2 | 0.55 | 0.8 |
| M4-M1 | 0.68 | 0.81 |
| M4-M2 | 0.29 | 0.71 |
| M4-M3 | 0.3 | 0.75 |






**Table 4.** Average Apr-Oct epilimnion depth estimates (m) derived using high resolution and low resolution modelled water temperature data, and the difference calculated between the high resolution and low resolution estimate (m), for Lough Feeagh and Lake Erken.

| Method | Lough Feeagh | | | Lake Erken | | |
|---|---|---|---|---|---|---|
| | High resolution average (m) | Low resolution average (m) | Difference (m) | High resolution average (m) | Low resolution average (m) | Difference (m) |
| M1 | -22.1 | -22.2 | 0.1 | -10.9 | -10.9 | 0 |
| M2 | -31.7 | -34.9 | 3.2 | -11.9 | -13.1 | 1.2 |
| M3 | -32.0 | -35.2 | 3.2 | -12.1 | -13.1 | 1 |
| M4 | -22.1 | -22.6 | 0.5 | -11.3 | -11.5 | 0.2 |

**Table 5.** Average difference in average Apr-Oct epilimnion depth estimates (m) between each profile based method (M1-M4) calculated using lake modelled data and the modelled turbulence method (M5) and average of all Pearson's correlation coefficients (r) calculated for each profile-based method and M5 (e.g. for all threshold combinations between M5 and M1 etc.), for Lough Feeagh and Lake Erken. Positive average differences indicate that the modelled turbulence method was shallower.

| Method | Lough Feeagh | | Lake Erken | |
|---|---|---|---|---|
| | Difference in average epilimnion depth (m) | r | Difference in average epilimnion depth (m) | r |
| M5-M1 | 1.3 | 0.90 | 0.0 | 0.89 |
| M5-M2 | 11.0 | 0.55 | 1.0 | 0.73 |
| M5-M3 | 11.2 | 0.54 | 1.1 | 0.72 |
| M5-M4 | 1.3 | 0.88 | 0.4 | 0.85 |



**Figures**

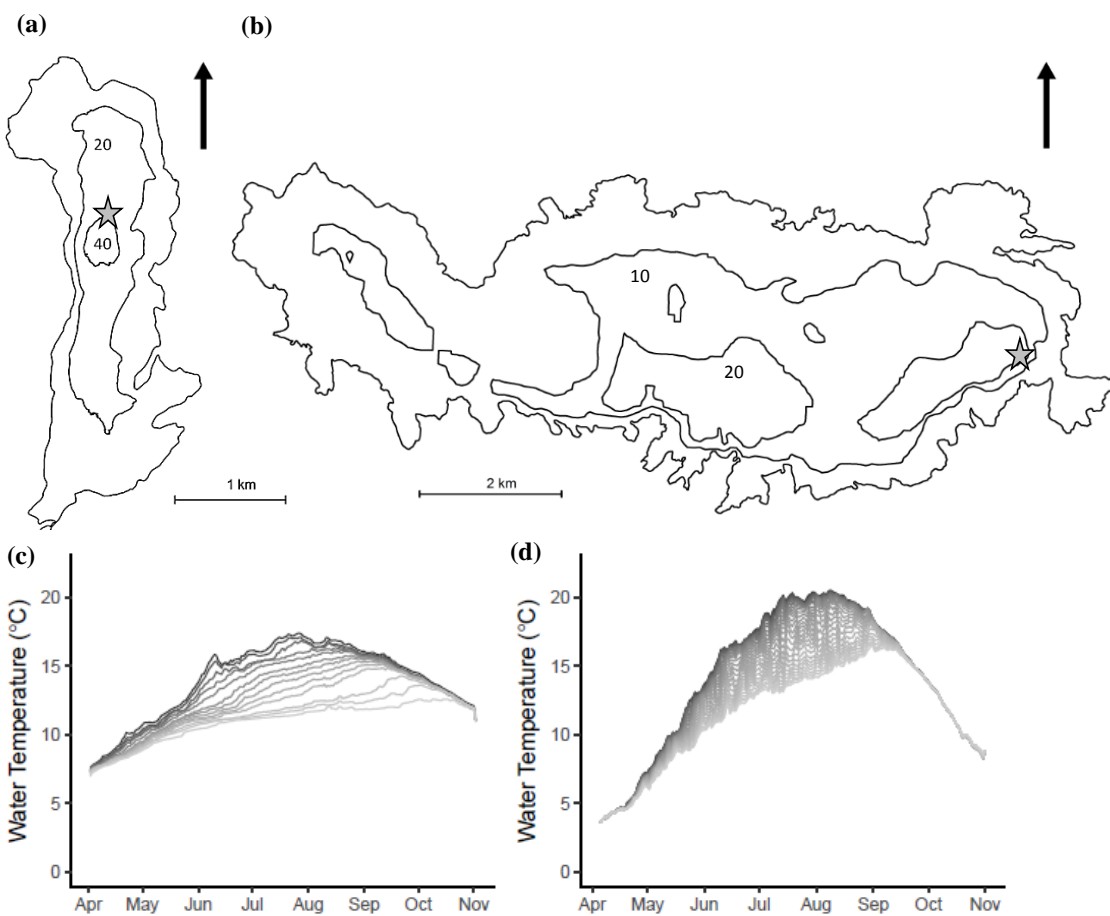

**Figure 1.** Bathymetric map of Lough Feeagh in Ireland **(a)** and Lake Erken in Sweden **(b)**, where the grey stars shows the locations of the automatic monitoring buoys used for measuring high-frequency water temperature profiles in both lakes, and long-term average water temperature for each Julian day for all measured depths, for Lough Feeagh (profiles = 2778, years = 10) **(c)** and Lake Erken (profiles = 2005, years = 7) **(d)**.

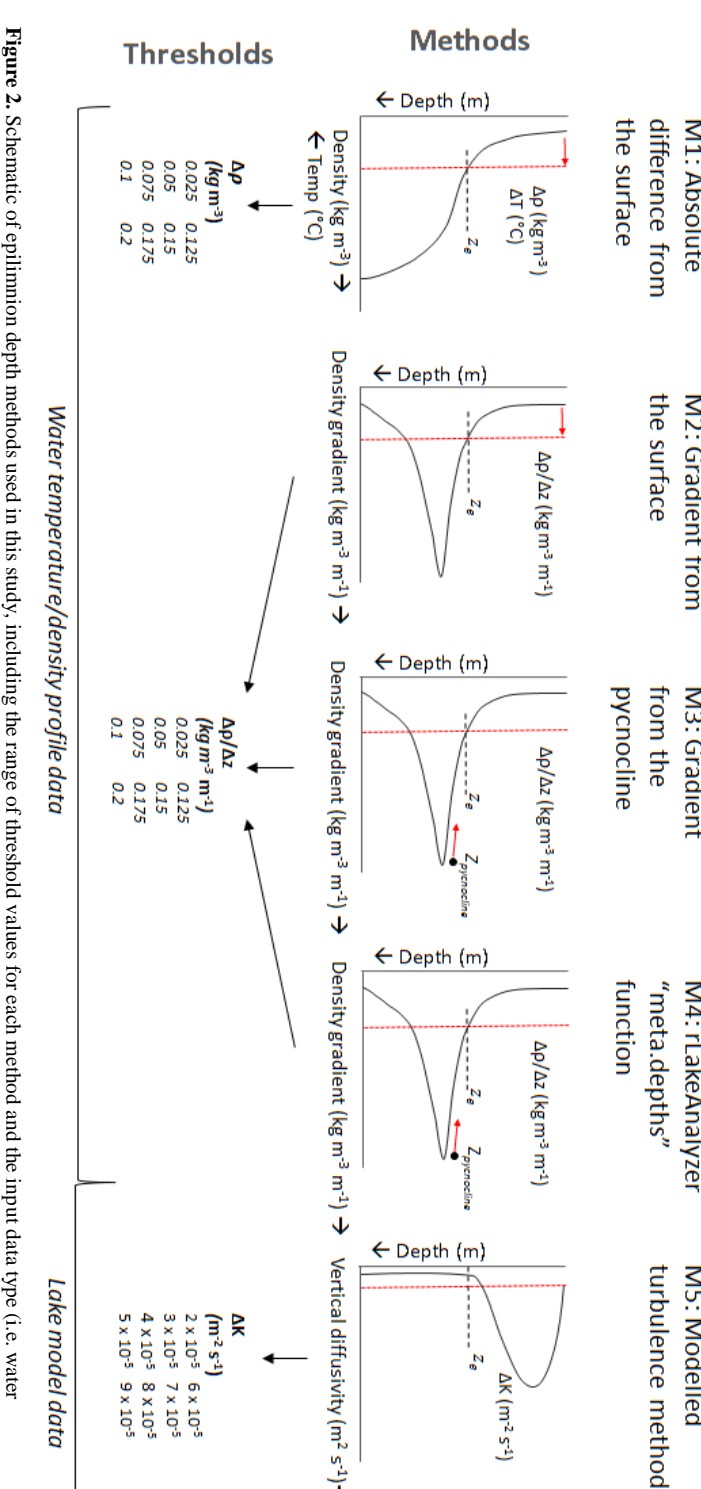

**Figure 2.** Schematic of epilimnion depth methods used in this study, including the range of threshold values for each method and the input data type (i.e. water temperature/density profile data or lake modelled data).



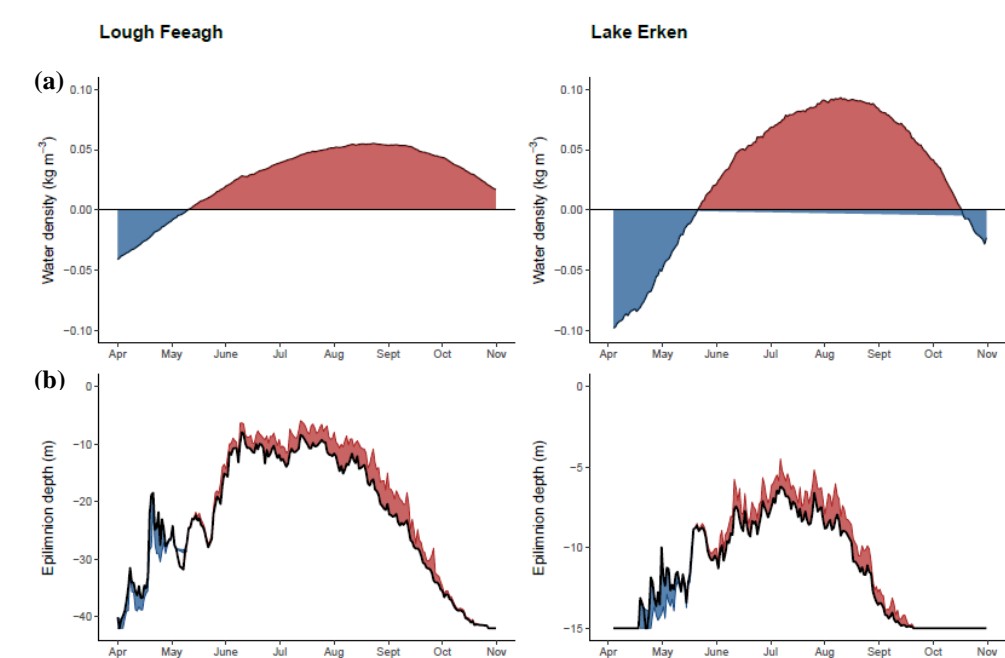

**Figure 3.** Long-term average for each Julian day of the difference in water density (kg m$^{-3}$) induced by an
increase of 1 °C in water temperature, relative to 0.1 kg m$^{-3}$ (black line with the red shaded area demonstrating
when the change induced by an increase of 1 °C change was greater than 0.1 kg m$^{-3}$ and the blue shaded area for
10  when it was less than 0.1 kg m$^{-3}$) **(a)**, and the long-term average for each Julian day of epilimnion depth
calculated using a water temperature threshold of 1 °C (the black line), compared to a water density threshold of
0.1 kg m$^{-3}$ (shaded area, with the red shaded areas demonstrating when water density estimates were shallower
and the blue shaded area for when they were deeper) **(b)**, for Lough Feeagh and Lake Erken.

30

35

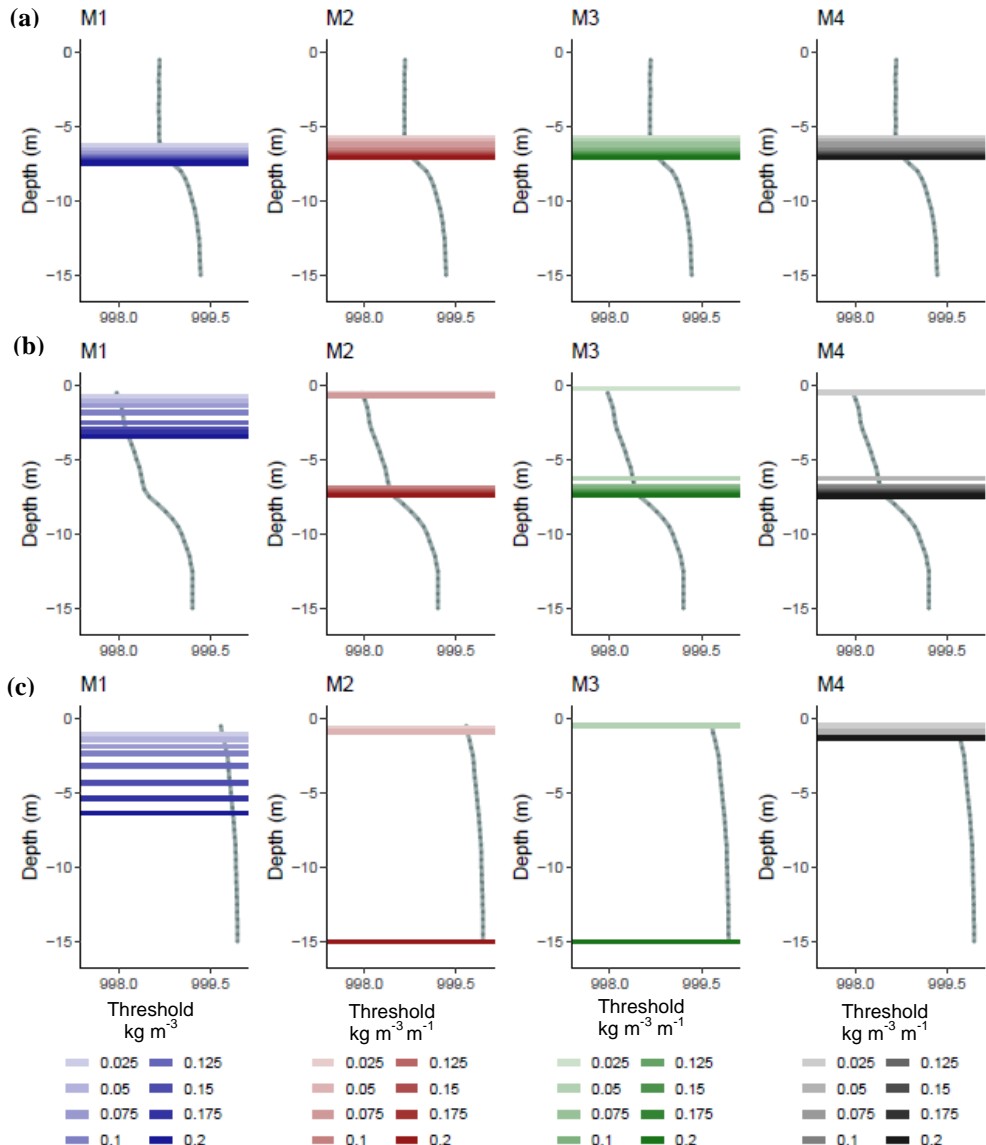

**Figure 4.** An example of water column profiles with the epilimnion depth estimates superimposed (horizontal lines) for all for all profile-based epilimnion depth methods calculated using the full range of thresholds for each. The water columns can be categorised as a three-layered water column structure **(a)**, an intensely stratified profile **(b)**, and a near-isothermal profile **(c)**, all from Lake Erken only.



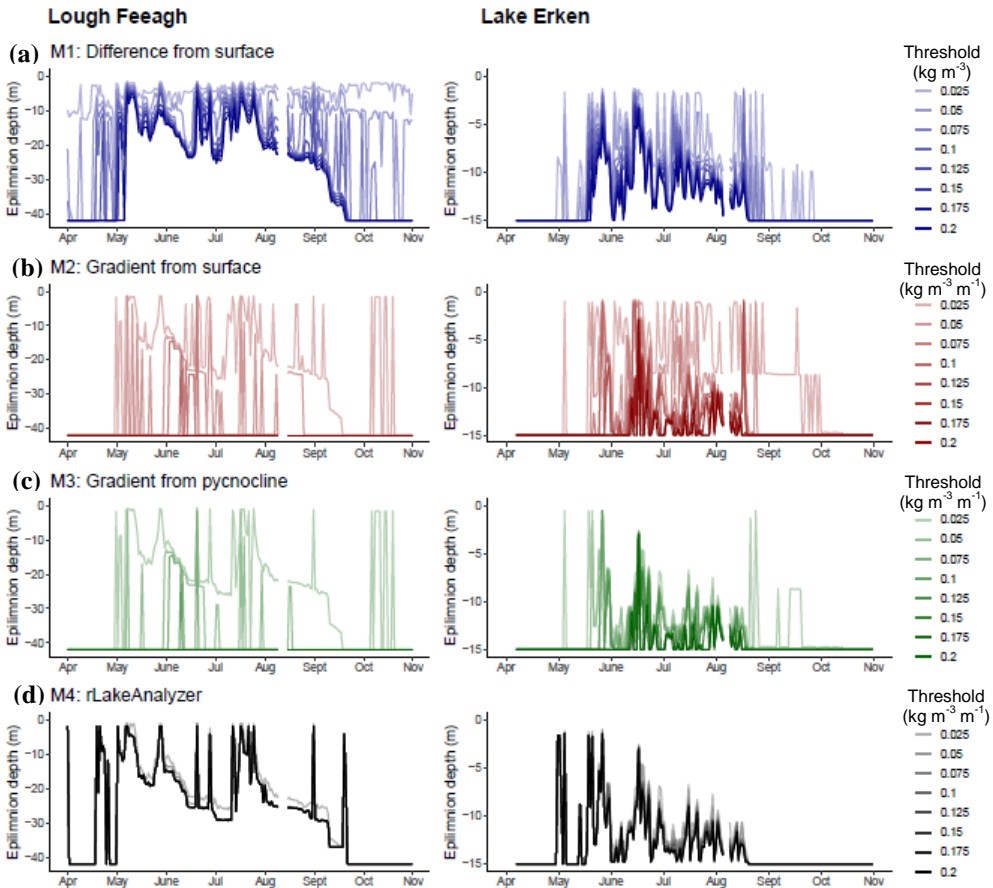

**Figure 5**. Daily epilimnion depth estimates using measured data for 2017 from Lough Feeagh and Lake Erken, showing estimates from all profile-based epilimnion depth methods, including M1, the absolute difference from the surface method **(a)**, M2, the gradient from the surface method **(b)**, M3, the gradient from the pycnocline method **(c)** and M4, the rLakeAnalyzer method **(d)**, calculated using the full range of thresholds, and for each lake.





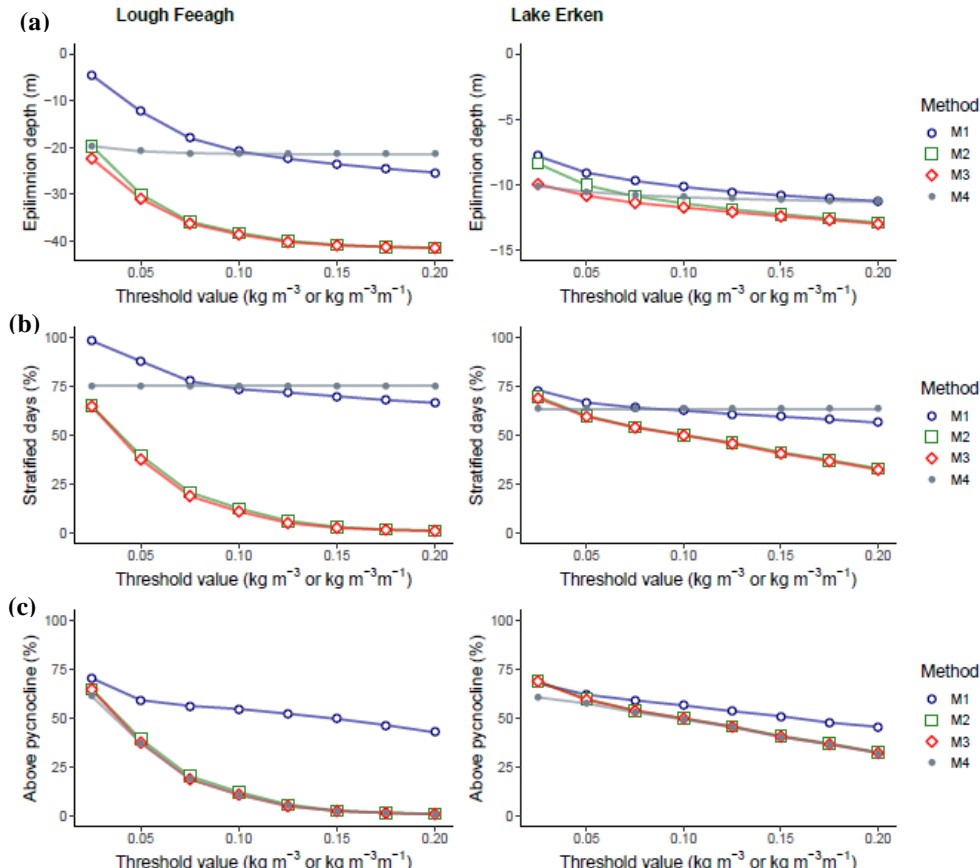

**Figure 6.** Average Apr-Oct epilimnion depth (m) **(a)**, percentage of stratified days, defined as days where the epilimnion depth was identified at a depth greater than the lake maximum measured depth (%) (where a larger percentage value indicated a higher occurrence of days identified as stratified) **(b)**, and percentage of days where the epilimnion depth was above the pycnocline, defined as the number of days where the epilimnion was identified at a depth shallower than the maximum density gradient (where a larger percentage value indicated a lower occurrence of days erroneously extending into the metalimnion) **(c)**, for Lough Feeagh and Lake Erken.

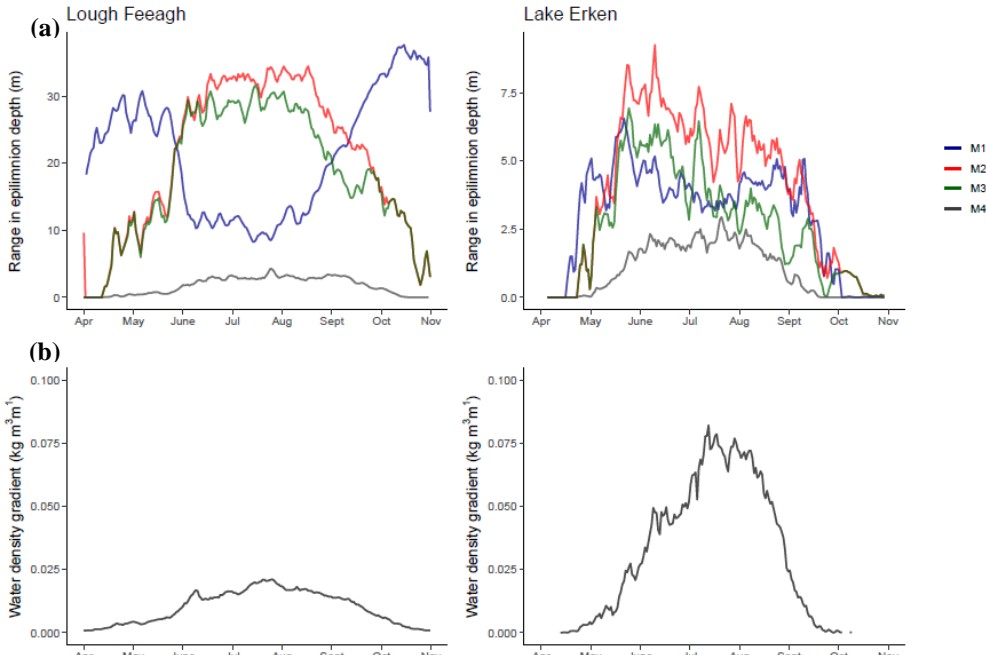

105    **Figure 7.** Range between the shallowest and deepest estimate for each method calculated from long-term daily average epilimnion depth estimates for each Julian day, where a large range suggests high threshold sensitivity and a small range suggests low sensitivity **(a)**, and long-term daily average water density gradient, calculated based on the surface and maximum measured depths **(b)**, for Lough Feeagh and Lake Erken.