# Peer review of "Variability in epilimnion depth estimations in lakes"

_Hydrology and Earth System Sciences, 2020_

## Referee Comment (RC1) · Anonymous Referee #1 · 7 Jul 2020

**General comments**

The article evaluates the epilimnion depth estimate in high-frequency data made by four different methods, including the effect of defining different thresholds that these methods require the user to choose. It also made this estimate using a hydrodynamic physical model. The aim of the study was to highlight the variability of the epilimnion depth estimates and how this variability could impact inferences about lake processes. This study draws attention to the need for researchers to unify their consensus on this topic, allowing comparisons between the results of different studies.

This is a very important study that addresses a hot topic in limnology and oceanography. With the increase in studies evaluating different bodies of water and the ease of obtaining increasingly sensitive measuring equipment, with the ability to perform and

store an increasing number of measurements, we have entered the era of Big Data. With the increasing availability of data, the need for models capable of extracting the correct information from them also increases. The correct adjustment of these models depends on studies of this type.

I believe that the result obtained and discussed in the article mixed the estimates made in clearly stratified water columns, with estimates made in weakly stratified water columns with estimates made in water columns with the presence of multiple stratifications. Therefore, the described variability is not only due to the distinction between methods and limits, but also to the application of the methods under different conditions, and this should be clearer.

Specific comments

P. 8 L. 281-283: I think that authors should make a clearer distinction between primary and secondary thermocline. For example, the method described by Read et al (2011) allows us to estimate the two thermoclines and, therefore, is more sensitive than the other methods, which do not consider this possibility. The comparison between the methods must consider the presence of these superficial micro-stratifications. For example, in the graphs b) of figure 4, there is clearly the presence of these micro-stratifications that are hampering some methods of identifying the main thermocline. In other words, it is not fair to fit a model that expects only one stratification with profiles that show various stratifications. It is obvious that the estimate will not be satisfactory. It is extremely necessary to make a pre-filter, removing the superficial layers of values before the estimate is made. This method was applied by Pujoni et al. (2019).

P. 8 L. 288-289: In this same line, we must discuss and define a threshold of what we call "homogeneous water column". I don't think it makes sense to compare the methods using profiles with low stability of the water column. If we no longer have a clear stratification, the methods should not be applied, as they will look for a thermocline that does not exist. I may be wrong, but the water column in the graphs c) in figure 4 is

homogeneous and should not be subjected to comparison with these models.

P. 9 L. 350: Why did the authors use the range to estimate variability? The range is sensitive to outliers. Why not use standard deviation, which is a more robust estimate of variation?

P. 10 L. 379-382: I would suggest showing some graphs of density profiles with the estimated depths of the methods so that it would be easy to see why there were such differences and whether one method made a "better" estimate than the other.

I would suggest that the authors discuss a little about the visual assessment of profiles. Should we rely on this visual assessment to try to "correct" the biased estimates made by the models?

---

## Referee Comment (RC2) · Anonymous Referee #2 · 10 Jul 2020

The study addresses one of the fundamental paradigms of limnology, the three-layer structure of the stratified water column. Here, the authors compare different algorithms to quantify the depth of the epilimnion, defined as a well-mixed, homogenous surface layer. As a general and agreed on mathematical definition of the distinction between epilimnion, metalimnion and hypolimnion is missing, different arbitrary thresholds for the epilimnion depth were investigated on two lake systems. As this paper aims to quantify the variability of epilimnion depth estimations and the methodological differences between alternative algorithms, it is of huge interest for a wide audience of limnologists, water managers, oceanographers, modelers, and environmental engineers. The study design, methods and results are well explained, although some paragraphs should be improved. Overall, the results of the study are important for future limnological research and are challenging our current conceptual paradigm.

General points:

- Quantifying the data variability by computing the range has potential shortcomings, e.g. bias by outliers, no information regarding the distribution of data. Specifically, in this study the authors did state the maximum and minimum values enabling every reader to calculate the range themselves. The authors should think about computing and stating alternative metrics like the variance, standard deviation and/or the inter-quantile range.

- The manuscript stresses that the data used were sampled on a high frequency (1-2 min), but I do not see the advantage of using high-frequency data here in this study as diurnal epilimnion depth trends/oscillations were not discussed at all. At the end, as mostly seasonal averages, ranges or data point fluctuations were discussed, short-term dynamics were neglected. Would a study that uses for example bi-weekly sampled vertical profiles over 50 years give the same results?

- I'd advise the authors to discuss the challenge of multiple pycnoclines by microstratification more intensively in the manuscript. The occurrence of these profiles especially in Erken, which seems to behave more polymictic than Feeagh, is more interesting to real-world applications (and which method should be used then) than discussing shortcomings of the classical three-layer structure.

- Just to make future studies more concise, it would be better to investigate lakes whose monitoring programs did include vertical temperature, density and velocity profiles, e.g. estimating eddy diffusivities by ADCP. The addition of GOTM to this study to estimate turbulence seems a bit half-hearted as the calibration-validation was not described nor any figure showing the M5 results included in the manuscript.

Specific points: - L15-16: I'd recommend dropping the quotation marks around epilimnion and metalimnion in the abstract as it's a bit confusing to the reader. Further,

as the first lines discuss the three-layered structure, I'd recommend shortly mentioning the hypolimnion here.

- L23: If needed you could also exchange 'approaches' and 'methods' with 'algorithms' throughout the manuscript.

- L28: The phrase 'complex water column structures' is a bit confusing here. Are you referring to cases when the three-layer paradigm is violated? Maybe rephrase to complex thermal water column structure?

- L35: The phrase 'less problematic' is quite vague, do you mean 'introduces less bias'?

- L40: What do you mean by 'rapid gradients'? I'd recommend 'steep gradients'

- L48-50: I'd recommend stressing these important statements more throughout the discussion at the end

- L74-75: Seems like sentences L48-50 already explained that in reality there are no exact cut-offs, so how could there even be a consistent method used throughout limnology?

- L86: Is the vertical turbulence profile referring to a profile of turbulent eddy diffusivities? Here, the authors could also discuss field methods which measure turbulence in lakes, e.g. through velocity loggers. Or methods estimating diffusivities from water temperature profiles, e.g. gradient flux method by Heinz et al (1990) or heat budget method by Jassby and Powell (1975)

- L116: I'd recommend moving the sentence "The lakes differ in many characteristics, [...]" to the beginning of the paragraph

- 2.3 Simulated data: Tab. 1 suggests that the model was calibrated and validated. Which time periods were used? In this paragraph, the investigated fit criteria should also be mentioned. Also, in line 153, was the parameter of the minimum turbulent kinetic energy calibrated or, alternatively, which parameters affecting the min. turbulent

kinetic energy were calibrated? This sentence is rather unclear.

- 2.5 Analysis methods: This paragraph would benefit from sub-headings to make it easier for the reader to follow (similar structure as the Results would be beneficial)

- L208: Here the colors are mentioned but the figure is not referenced. If you do not want to cross-reference the figure yet, maybe just write about 'color-coding'.

- L232: 'logical and numerical schemes' is unclear to me. What do you mean by logical scheme here?

- L248: Is there a reason why the sensor deployment sensitivity was not tested on monitored data (e.g. by removing some loggers)?

- 3.2 Comparison between water density based methods: In my opinion, this paragraph is a crucial finding of the study as it discusses how deviations from the three-layer paradigm affected the results of the algorithms. Can you state how many profiles/observations points were either a) (classical paradigm), b) (multiple pycnoclines) or c) (weakly stratified)? Were most profiles following the three-layer structure or is the complex case b) dominating?

- L301: Here it would be beneficial for the reader to state the average epilimnion depth per lake plus the variance and the quantiles.

- L330: Shouldn't it be '[...] epilimnion depth was identified at a depth above [...]' instead of "greater"? As everything is referenced to the surface, wouldn't greater mean deeper?

- L332: It seems there's a word missing here

- L336: Do you mean 0.025 kg/m3/m instead of 0.25 kg/m3/m which would be higher than the maximum investigated threshold?

- L360: The phrase "[...] M2 had typically a higher threshold range than M3 [...]" confuses me. Do you mean that the range between the thresholds was higher?

[Figure]

- 3.4 Sensitivity of epilimnion depth: Can you test if the differences between the averages were significant?

- L405: The phrase 'particularly distinct systematic difference' seems a bit excessive

- L406: Is '[...] was equivalent to using different threshold values [...]' referring to density threshold values?

- L424: Do you actually mean 'acceleration of epilimnion deepening' instead of shallowing here? Shouldn't the epilimnion deepen relative to the surface height during stratification onset?

- L429: But these methods are detecting a layer that is specifically not isothermal relative to themselves, right?

- L448: The phrase '[...] not be suitable for use with water density metric [...]' is unclear to me. Which water density metric are you talking about?

- L453-466: A figure showing these results would be beneficial for the reader, or how M5 compares to the other models.

- L465: Are profile data here referring to water temperature profile data?

- L515: I'd recommend not to write about 'problematic' here, maybe 'less bias', 'conservative assumption'?

- Table 3: Could you exchange the table with either a correlation matrix or correlation plot?

- Fig. 2 is great, I like it a lot! It makes the whole study easier to understand.

- Fig 3.: Why is there a thin blue shaded area below the red shade in Lake Erken?

---

## Author Comment (AC1) · 3 Sep 2020

**Author response to Referee #1 comments for Manuscript #HESS-2020-222, "Variability in epilimnion depth estimations in lakes" by Harriet L. Wilson, Ana I. Ayala, Ian D. Jones, Alec Rolston, Don Pierson, Elvira de Eyto, Hans-Peter Grossart, Marie-Elodie Perga, R. Iestyn Woolway, Eleanor Jennings.**

Thank you for your valuable comments which have enabled us to improve our study. Please find below our responses to each comment respectively. Updated text is in italics.

**1. Referee comment:** The article evaluates the epilimnion depth estimate in high-frequency data made by four different methods, including the effect of defining different thresholds that these methods require the user to choose. It also made this estimate using a hydrodynamic physical model. The aim of the study was to highlight the variability of the epilimnion depth estimates and how this variability could impact inferences about lake processes. This study draws attention to the need for researchers to unify their consensus on this topic, allowing comparisons between the results of different studies.

This is a very important study that addresses a hot topic in limnology and oceanography. With the increase in studies evaluating different bodies of water and the ease of obtaining increasingly sensitive measuring equipment, with the ability to perform and store an increasing number of measurements, we have entered the era of Big Data. With the increasing availability of data, the need for models capable of extracting the correct information from them also increases. The correct adjustment of these models depends on studies of this type.

**Author response:** Thank you for these positive comments. We greatly appreciate the enthusiasm for the study as a topic of current relevance.

**2. Referee comment:** I believe that the result obtained and discussed in the article mixed the estimates made in clearly stratified water columns, with estimates made in weakly stratified water columns with estimates made in water columns with the presence of multiple stratifications. Therefore, the described variability is not only due to the distinction between methods and limits, but also to the application of the methods under different conditions, and this should be clearer.

**Author response:** This is an important point by the reviewer, and we agree that both 1) the presence of multiple stratifications and 2) weakly stratified profiles, contribute to the large variability found between epilimnion depth estimates. In fact, it was *particularly* when water column profiles did not conform to the idealised three-layered structure, that we found the greatest divergence between epilimnion depth methods. However, we would argue it is not simply a matter of filtering out weakly stratified or multiple stratification profiles, since this would result in large data gaps, including in peak summer, and is limited by the same issues of subjectivity as the definition of the epilimnion depth itself. Instead, there is a need for users to acknowledge both the systematic differences between methods, and the application of the methods under different conditions, in respect to their specific purpose. We have addressed this point in the revised manuscript (please see Comment #3 for relevant edits to text).

**3. Referee comment**: P. 8 L. 281-283: I think that authors should make a clearer distinction between primary and secondary thermocline. For example, the method described by Read et al (2011) allows us to estimate the two thermoclines and, therefore, is more sensitive than the other methods, which do not consider this possibility. The comparison between the methods must consider the presence of these superficial micro-stratifications. For example, in the graphs b) of figure 4, there is clearly the presence of these microstratifications that are hampering some methods of identifying the main thermocline. In other words, it is not fair to fit a model that expects only one stratification with profiles that show various stratifications. It is obvious that the estimate will not be satisfactory. It is extremely necessary to make a pre-filter, removing the superficial layers of values before the estimate is made. This method was applied by Pujoni et al. (2019).

**Author response:** See the response to Comment #2 above and Comment #4 which are related. We do not think it is appropriate in this study to make a defined distinction between primary and secondary thermoclines, or filter/smooth out these stratifications for the following reasons; 1) determination of primary/secondary thermoclines is not objective and the selection of profiles requiring smoothing would demand further arbitrary thresholds (instead we argue that this topic deserves a separate study, investigating the full implications of different approaches), 2) in this study we aimed to estimate variability in epilimnion depth estimates between *common and existing* methods (rather than introduce new methods), where issues related to microstratification are a result/discussion point, 3) smoothing of these microstratifications may not be suitable for some applications, particularly where it is assumed that the epilimnion is isothermal and well-mixed. Nevertheless, we do fundamentally agree with the reviewer that this is an important issue that should be clearer in the text, therefore we propose editing the discussion to emphasise these points (see below). We also now cite the work of Pujoni et al., (2019) and Read et al., (2011) in this context.

L.388: '*The concept of the epilimnion, and more widely, the three-layered structure of a stratified lake, is fundamental in limnology. Yet, despite the ubiquity of the term, there is no objective or generic approach for defining the epilimnion and a diverse number of approaches prevail in the literature. In a comprehensive analysis of high-frequency, multi-year data from two lakes, this study has highlighted the extent to which common water temperature profile based epilimnion depth estimates differ. The level of variability in epilimnion depth estimates calculated using common methods and threshold values, was exceedingly high. This result calls into question the practice of arbitrary method selection and comparing findings between studies which use different methods or even just different thresholds. The magnitude of variability also casts ambiguity on the calculation of key biogeochemical and ecological processes in a lake that rest on the assumption that the layers of a lake are well defined, including calculations of metabolic rates, and oxygen fluxes (e.g. Coloso et al., 2008, Foley et al., 2012, Obrador et al., 2014, Winslow et al., 2016).*

*In an idealised stratified profile, the epilimnion is portrayed as near-uniform in water temperature or density and clearly delineated from a well-defined metalimnion. However, many measured profiles, at least within this study, did not conform to this idealised three-layered structure. Instead the water columns were often more complex, including multiple pycnoclines and near-surface micro-stratification layers, or the boundaries between the epilimnion/metalimnion were blurred. One approach to this issue is to filter out appropriate water column profiles or apply functions that coerce the profile into the expected structure (Read et al., 2011, Pujoni et al., 2019, Gray et al., 2020). Filters, additional conditions or smoothing functions, however, may suffer from many of the same challenges as the estimation of the epilimnion depth, since they attempt to discretise data based on arbitrary criteria (Kraemer et al., 2020). For example, our analysis of temporally high resolution time series data emphasised that rather than jumping from states, such as stratified or isothermal, changes in the water*

*column occurred over an evolving continuum and often fluctuated between states. Similarly, the distinction between additional layers, such as the primary or secondary pycnocline, is fraught with the same issues of arbitrariness as discussed (Read et al., 2011). This study demonstrates that when epilimnion depth estimation methods, which are theorised for a three-layered water column, are applied to non-conforming water columns, they diverge widely on the location of the epilimnion depth, and at times, may not even be underpinned by the same theoretical assumptions. Since none of these methods can be considered the 'true' definition of the epilimnion depth, it is necessary to understand the degree to which methods differ. Improved understanding of their systematic differences will facilitate the use of methods that appropriately capture different processes, such as, air-water exchanges, thermocline entrainment or suspension of materials. Due to the realised complexities of observed and aggregated profile data, we may benefit from new approaches to water column discretisation that incorporate the vast proportion of profiles which do not conform neatly to the three-layered paradigm.'*

**4. Referee comment:** P. 8 L. 288-289: In this same line, we must discuss and define a threshold of what we call "homogeneous water column". I don't think it makes sense to compare the methods using profiles with low stability of the water column. If we no longer have a clear stratification, the methods should not be applied, as they will look for a thermocline that does not exist. I may be wrong, but the water column in the graphs c) in figure 4 is homogeneous and should not be subjected to comparison with these models.

**Author response:** This is another great point by the reviewer, and we agree that the stability of the water column has an influence on the results. Inherently, however, there are conditions within each of the methods, relating to the degree of stratification required to estimate the epilimnion depth. Method 1 has the precondition that the range in water density must be greater than the threshold value, else the epilimnion depth is assigned to the maximum lake depth. Similarly, Methods 2 and 3 have the precondition that the water density gradient must be greater than the threshold value. Finally, Method 4, the rLakeAnalyzer, is slightly different as it initially filters out profiles based on a 1°C water column range and will then identify the maximum density gradient regardless of the threshold value. As discussed inComment #3, a further stability-based condition could be introduced, but would suffer from being an arbitrary threshold to which the rest of the results would become dependent. Again, it is the fact that studies are currently using different approaches with different inherent stability thresholds that can contribute to the confusion caused when comparing studies, and this is one of the key points we are raising in this manuscript.

Note, though, that we did extensively investigate the use of pre-filters, including top-bottom density differences, water column total density gradient and Schmidt stability values. However, echoing the presented findings of the epilimnion depth analysis, we found different methods and thresholds largely altered the period that was deemed stratified. This analysis could not be justly presented in this study, without overly complicating the manuscript. In addition, they resulted in the removal of large amounts of data, even within peak summer, which is not suitable for analysis of mixing events for example. Finally, presenting the full time series results demonstrated the perils of using temporal means of epilimnion depth. For example, for calculating the summer mean epilimnion depth, the un-filtered mean will be influenced by periods of very deep epilimnion depth estimates (i.e. when the water column is nearly isothermal), while the filtered mean would not be representative of conditions during the full summer, but rather a subset of the stratified profiles.

**5. Referee comment:** P. 9 L. 350: Why did the authors use the range to estimate variability? The range is sensitive to outliers. Why not use standard deviation, which is a more robust estimate of variation?

**Author response:** The reviewer has highlighted an important point and in the revised document we will use inter-quartile range which more robust to outliers than range. To address this we update the methods, results (Table 2 and Figure 7), although the findings are very similar.

**Table 2.** Summary of statistics for each method, showing the mean (m), minimum (i.e. shallowest estimate) (m), maximum (i.e. deepest estimate) (m) and the interquartile range (m) of the April-October epilimnion depth estimates (summarised from the results shown in Fig.6a), and the mean Pearson's correlation coefficient (r) for each method, representing the mean correlation for all possible combinations between threshold values., for Lough Feeagh and Lake Erken.

| Method | Lough Feeagh | | | | | Lake Eken | | | | |
|---|---|---|---|---|---|---|---|---|---|---|
| | Mean (m) | Min (m) | Max (m) | IQR (m) | r | Mean (m) | Min (m) | Max (m) | IQR (m) | r |
| M1 | -19.0 | -4.6 | -25.4 | 7.3 | 0.77 | -10.0 | -7.8 | -11.2 | 1.3 | 0.92 |
| M2 | -35.9 | -19.7 | -41.4 | 6.5 | 0.48 | -11.3 | -8.4 | -12.9 | 1.7 | 0.78 |
| M3 | -36.5 | -22.4 | -41.5 | 6.1 | 0.49 | -11.8 | -10.0 | -13.0 | 1.2 | 0.82 |
| M4 | -21.1 | -19.7 | -21.5 | 0.3 | 1.00 | -11.9 | -10.1 | -11.3 | 0.5 | 0.99 |

[Figure]

**Figure 7.** Inter-quartile range between the shallowest and deepest estimate for each method calculated from long-term daily mean epilimnion depth estimates for each Julian day, where a large range suggests high threshold sensitivity and a small range suggests low sensitivity **(a)**, and long-term daily mean water density gradient, calculated based on the surface and maximum measured depths **(b)**, for Lough Feeagh and Lake Erken.

L. 225-227: *'We also summarised these statistics for each method, showing the mean, minimum (shallowest), maximum (deepest) and interquartile range for each method, to demonstrate differences between methods. A large interquartile range in epilimnion depth estimates, indicated high sensitivity to the threshold value.'*

L. 317 – 323: *'For both lakes, the interquartile range in the mean Apr-Oct epilimnion depth estimates for each method was very high for M2, M1 and M3, indicating high threshold sensitivity in these methods. Method M4 had a substantially lower interquartile range than all other methods and a very high mean Pearson's correlation coefficient, indicating that both the mean value and the temporal pattern of the epilimnion depth were only weakly influenced by the threshold value. In both lakes, methods M2 and M1, where the epilimnion depth was defined from the surface downwards, had a higher interquartile range in estimates calculated with different threshold values, compared to methods M3 and M4, where the epilimnion was defined from the pycnocline upwards.'*

L.348 – 366: '*For all methods, threshold sensitivity fluctuated seasonally, although varied in pattern (Fig. 7). Threshold sensitivity was shown by the interquartile range between the shallowest and deepest epilimnion depth estimates calculated for all threshold values. In Lough Feeagh, M1 had a smaller range in epilimnion depth estimates during the peak summer months of June, July and August, compared with months when the onset and overturn of stratification commonly occurred. During periods of transient stratification, the stability of the water column was often low but frequent changes in the near-surface water density, induced large differences between estimates calculated using small thresholds compared with large threshold values. In contrast, methods M2 and M3 had the highest range in estimates occurring during the peak summer months. Even during peak summer in Lough Feeagh, gradients in the water column were relatively small (Fig. 7b), which resulted in a very large range between the smallest threshold values which found a near-surface epilimnion depth, and the largest thresholds that often found no epilimnion depth at all, therefore defaulting to the deepest depth. In Lake Erken, the water density gradients were typically much larger, and methods M1, M2 and M3 all peaked during May and June, when gradients in the water column were typically increasing but prone to fluctuations. For both lakes, M2 had typically a higher threshold interquartile range than M3 during peak summer and the overturn period, which was related to the common development of a secondary pycnocline. M4 produced much lower interquartile ranges in the epilimnion depth throughout the year, since as long as the 'mixed.cutoff' filter was met, the epilimnion depth was defaulted to the pycnocline if the threshold was not exceeded, thus largely reducing the ability for large differences to occur. The interquartile range in epilimnion depth estimates for M4 was highest during the peak summer months, which was when the epilimnion depth was typically shallowest and more frequently defined by the threshold value rather than defaulting to the pycnocline.'*

**6. Referee comment:** P. 10 L. 379-382: I would suggest showing some graphs of density profiles with the estimated depths of the methods so that it would be easy to see why there were such differences and whether one method made a "better" estimate than the other. I would suggest that the authors discuss a little about the visual assessment of profiles. Should we rely on this visual assessment to try to "correct" the biased estimates made by the models?

**Author response:** Visual assessment of profiles is certainly very helpful and is also commonly practised in limnology. We think that with 5 tables and 7 figures it may be excessive to add additional figures. Our intention with Figure 4 was to demonstrate to the user all methods/thresholds on three distinctly different profiles, however, visual assessment is not part of our result analysis. The focus for this study is for using high-frequency data and multi-lake analysis, and therefore the goal is to find methods that can be used systematically without being tailored to specific lakes.

We are interested however in visual assessment of epilimnion depth and we have conducted a survey investigating where limnologists visually identify the epilimnion depth using profiles from anonymous lakes. This is something we would like to publish at a later date as a short discussion paper, but is not appropriate for automated analysis of high frequency data.

---

## Author Comment (AC2) · 3 Sep 2020

**Author response to Referee #2 comments for Manuscript # hess-2020-222, "Variability in epilimnion depth estimations in lakes" by Harriet L. Wilson, Ana I. Ayala, Ian D. Jones, Alec Rolston, Don Pierson, Elvira de Eyto, Hans-Peter Grossart, Marie-Elodie Perga, R. Iestyn Woolway, Eleanor Jennings.**

Thank you for your valuable comments which have enabled us to improve our study. Please find below our responses to each comment respectively. Updated text is in italics.

**1. Referee comment:** The study addresses one of the fundamental paradigms of limnology, the three-layer structure of the stratified water column. Here, the authors compare different algorithms to quantify the depth of the epilimnion, defined as a well-mixed, homogenous surface layer. As a general and agreed on mathematical definition of the distinction between epilimnion, metalimnion and hypolimnion is missing, different arbitrary thresholds for the epilimnion depth were investigated on two lake systems. As this paper aims to quantify the variability of epilimnion depth estimations and the methodological differences between alternative algorithms, it is of huge interest for a wide audience of limnologists, water managers, oceanographers, modelers, and environmental engineers. The study design, methods and results are well explained, although some paragraphs should be improved. Overall, the results of the study are important for future limnological research and are challenging our current conceptual paradigm.

**Author response:** Thank you for these positive comments. We greatly appreciate the enthusiasm for the study as a topic of current relevance.

**2. Referee comment:** Quantifying the data variability by computing the range has potential shortcomings, e.g. bias by outliers, no information regarding the distribution of data. Specifically, in this study the authors did state the maximum and minimum values enabling every reader to calculate the range themselves. The authors should think about computing and stating alternative metrics like the variance, standard deviation and/or the interquantile range.

**Author response:** The reviewer has highlighted an important point and in the revised document we will use inter-quartile range which more robust to outliers than range. To address this we update the methods, results (Table 2 and Figure 7), although the findings are very similar.

**Table 2.** Summary of statistics for each method, showing the mean (m), minimum (i.e. shallowest estimate) (m), maximum (i.e. deepest estimate) (m) and the interquartile range (m) of the April-October epilimnion depth estimates (summarised from the results shown in Fig.6a), and the mean Pearson's correlation coefficient (r) for each method, representing the mean correlation for all possible combinations between threshold values., for Lough Feeagh and Lake Erken.

| Method | Lough Feeagh | | | | | Lake Eken | | | | |
| --- | --- | --- | --- | --- | --- | --- | --- | --- | --- | --- |
| | Mean (m) | Min (m) | Max (m) | IQR (m) | r | Mean (m) | Min (m) | Max (m) | IQR (m) | r |
| M1 | -19.0 | -4.6 | -25.4 | 7.3 | 0.77 | -10.0 | -7.8 | -11.2 | 1.3 | 0.92 |
| M2 | -35.9 | -19.7 | -41.4 | 6.5 | 0.48 | -11.3 | -8.4 | -12.9 | 1.7 | 0.78 |
| M3 | -36.5 | -22.4 | -41.5 | 6.1 | 0.49 | -11.8 | -10.0 | -13.0 | 1.2 | 0.82 |
| M4 | -21.1 | -19.7 | -21.5 | 0.3 | 1.00 | -11.9 | -10.1 | -11.3 | 0.5 | 0.99 |

[Figure]

**Figure 7.** Inter-quartile range between the shallowest and deepest estimate for each method calculated from long-term daily mean epilimnion depth estimates for each Julian day, where a large range suggests high threshold sensitivity and a small range suggests low sensitivity **(a)**, and long-term daily mean water density gradient, calculated based on the surface and maximum measured depths **(b)**, for Lough Feeagh and Lake Erken.

L. 225-227: *'We also summarised these statistics for each method, showing the mean, minimum (shallowest), maximum (deepest) and interquartile range for each method, to demonstrate differences between methods. A large interquartile range in epilimnion depth estimates, indicated high sensitivity to the threshold value.'*

L. 317 – 323: *'For both lakes, the interquartile range in the mean Apr-Oct epilimnion depth estimates for each method was very high for M2, M1 and M3, indicating high threshold sensitivity in these methods. Method M4 had a substantially lower interquartile range than all other methods and a very high mean Pearson's correlation coefficient, indicating that both the mean value and the temporal pattern of the epilimnion depth were only weakly influenced by the threshold value. In both lakes, methods M2 and M1, where the epilimnion depth was defined from the surface downwards, had a higher interquartile range in estimates calculated with different threshold values, compared to methods M3 and M4, where the epilimnion was defined from the pycnocline upwards.'*

L.348 – 366: *'For all methods, threshold sensitivity fluctuated seasonally, although varied in pattern (Fig. 7). Threshold sensitivity was shown by the interquartile range between the shallowest and deepest epilimnion depth estimates calculated for all threshold values. In Lough Feeagh, M1 had a smaller range in epilimnion depth estimates during the peak summer months of June, July and August, compared with months when the onset and overturn of stratification commonly occurred. During periods of transient stratification, the stability of the water column was often low but frequent changes in the near-surface water density, induced large differences between estimates calculated using small thresholds compared with large threshold values. In contrast, methods M2 and M3 had the highest range in estimates occurring during the peak summer months. Even during peak summer in Lough Feeagh, gradients in the water column were relatively small (Fig. 7b), which*

*resulted in a very large range between the smallest threshold values which found a near-surface epilimnion depth, and the largest thresholds that often found no epilimnion depth at all, therefore defaulting to the deepest depth. In Lake Erken, the water density gradients were typically much larger, and methods M1, M2 and M3 all peaked during May and June, when gradients in the water column were typically increasing but prone to fluctuations. For both lakes, M2 had typically a higher threshold interquartile range than M3 during peak summer and the overturn period, which was related to the common development of a secondary pycnocline. M4 produced much lower interquartile ranges in the epilimnion depth throughout the year, since as long as the 'mixed.cutoff' filter was met, the epilimnion depth was defaulted to the pycnocline if the threshold was not exceeded, thus largely reducing the ability for large differences to occur. The interquartile range in epilimnion depth estimates for M4 was highest during the peak summer months, which was when the epilimnion depth was typically shallowest and more frequently defined by the threshold value rather than defaulting to the pycnocline.'*

**3. Referee comment:** The manuscript stresses that the data used were sampled on a high frequency (1-2 min), but I do not see the advantage of using high-frequency data here in this study as diurnal epilimnion depth trends/oscillations were not discussed at all. At the end, as mostly seasonal means, ranges or data point fluctuations were discussed, short-term dynamics were neglected. Would a study that uses for example bi-weekly sampled vertical profiles over 50 years give the same results?

**Author response:** We used sub-daily data to aggregate to daily means, and then conducted the analysis on these daily values. There are, in fact, a number of extra (and interesting) complications when considering mixed depth on sub-daily timescales, such as seiche activity or diel cycles, which we believe would be worth addressing, but would require a whole new study. However, there are still important advantages of using high-frequency data even at the daily time-step. These are; 1) Daily water temperature estimates are still high-frequency (compared to fortnightly or monthly monitoring campaigns) and are commonly collected or calculated within models, thus making analysis particularly relevant, 2) Daily estimates tell us a lot more about the temporal evolution of stratification patterns than fortnightly or monthly one-off profiles, which is an important consideration in our results, 3) As an aggregate of finer resolute data (i.e. mean of sub-daily measurements), the daily data we use represents the variability within the day and are not affected by the time of detection as manually collected profiles would be, 4) Daily data produces many profiles to analyse (i.e. if we were using, say, monthly data, we would need to have 30 times as many years of data to get the same number of profiles), which allows us to be more confident in our results (i.e. more robust mean epilimnion depth, more validation in the variability we report and more robust in comparing the methods). To demonstrate this to the reader, we propose changing the following line in the introduction.

*L.95: 'Although lower temporal resolution data is sufficient for investigating seasonal patterns, high-frequency data can be used to gain information on the level of day-to-day variability in epilimnion depth and demonstrates how methods perform over a continuum of water column conditions. In addition, through the vast number of measured profiles, high-frequency data offers a more robust comparison of methods, than previously demonstrated with manually collected datasets, and even when aggregated to the daily time-step is more representative of the sub-daily variability (Marcé et al., 2016).'*

**4. Referee comment:** I'd advise the authors to discuss the challenge of multiple pycnoclines by microstratification more intensively in the manuscript. The occurrence of these profiles especially in Erken, which seems to behave more polymictic than Feeagh, is more interesting to real-world

applications (and which method should be used then) than discussing shortcomings of the classical three-layer structure.

**Author response:** See the response to Comment #11 and Comment #20 which are related. We do not think it is appropriate in this study to make a defined distinction between primary and secondary thermoclines, or pre-filter out these superficial stratifications for the following reasons; 1) determination of primary/secondary thermoclines is not objective and the selection of profiles requiring smoothing would demand further arbitrary thresholds (instead this topic deserves a separate study to investigate the full implications of different approaches), 2) in this study we aimed to estimate variability in epilimnion depth estimates between *common and existing* methods (rather than introduce new methods), which highlighted the issue of micro-stratifications as enhancing variability, 3) smoothing of these micro-stratifications may not be suitable for some applications, particularly where it is assumed that the epilimnion is isothermal and well-mixed**.** However we do fundamentally agree that this is an important issue and would therefore intend to make the following additions/edits:

L.388: '*The concept of the epilimnion, and more widely, the three-layered structure of a stratified lake, is fundamental in limnology. Yet, despite the ubiquity of the term, there is no objective or generic approach for defining the epilimnion and a diverse number of approaches prevail in the literature. In a comprehensive analysis of high-frequency, multi-year data from two lakes, this study has highlighted the extent to which common water temperature profile based epilimnion depth estimates differ. The level of variability in epilimnion depth estimates calculated using common methods and threshold values, was exceedingly high. This result calls into question the practice of arbitrary method selection and comparing findings between studies which use different methods or even just different thresholds. The magnitude of variability also casts ambiguity on the calculation of key biogeochemical and ecological processes in a lake that rest on the assumption that the layers of a lake are well defined, including calculations of metabolic rates, and oxygen fluxes (e.g. Coloso et al., 2008, Foley et al., 2012, Obrador et al., 2014, Winslow et al., 2016).*

*In an idealised stratified profile, the epilimnion is portrayed as near-uniform in water temperature or density and clearly delineated from a well-defined metalimnion. However, many measured profiles, at least within this study, did not conform to this idealised three-layered structure. Instead the water columns were often more complex, including multiple pycnoclines and near-surface micro-stratification layers, or the boundaries between the epilimnion/metalimnion were blurred. One approach to this issue is to filter out appropriate water column profiles or apply functions that coerce the profile into the expected structure (Read et al., 2011, Pujoni et al., 2019, Gray et al., 2020). Filters, additional conditions or smoothing functions, however, may suffer from many of the same challenges as the estimation of the epilimnion depth, since they attempt to discretise data based on arbitrary criteria (Kraemer et al., 2020). For example, our analysis of temporally high resolution time series data emphasised that rather than jumping from states, such as stratified or isothermal, changes in the water column occurred over an evolving continuum and often fluctuated between states. Similarly, the distinction between additional layers, such as the primary or secondary pycnocline, is fraught with the same issues of arbitrariness as discussed (Read et al., 2011). This study demonstrates that when epilimnion depth estimation methods, which are theorised for a three-layered water column, are applied to non-conforming water columns, they diverge widely on the location of the epilimnion depth, and at times, may not even be underpinned by the same theoretical assumptions. Since none of these methods can be considered the 'true' definition of the epilimnion depth, it is necessary to understand the degree to which methods differ. Improved understanding of their systematic differences will facilitate the use of methods that appropriately capture different processes, such as, air-water exchanges, thermocline entrainment or suspension of materials. Due to the realised complexities of observed and aggregated profile data, we may benefit from new approaches to water column*

*discretisation that incorporate the vast proportion of profiles which do not conform neatly to the three-layered paradigm.'*

**5. Referee comment:** Just to make future studies more concise, it would be better to investigate lakes whose monitoring programs did include vertical temperature, density and velocity profiles, e.g. estimating eddy diffusivities by ADCP. The addition of GOTM to this study to estimate turbulence seems a bit half-hearted as the calibration-validation was not described nor any figure showing the M5 results included in the manuscript.

**Author response:** While there would be some advantages to analyse lakes with vertical eddy diffusivity data, vertical eddy diffusivity measurements are fairly rare and often taken only for short periods of time, or only taken at specific depths in the lake. Therefore, it is likely that epilimnion depth estimates will continue to be derived using water temperature profile data, unless sensor technology evolves dramatically. To address the comments on the modelled turbulence section we have added some figures to supplementary (see Comment #32) as well as additional information on the calibration-validation (see Comment #15), and edited the relevant paragraph in the discussion (Comment #11).

**6. Referee comment:** L15-16: I'd recommend dropping the quotation marks around epilimnion and metalimnion in the abstract as it's a bit confusing to the reader. Further, as the first lines discuss the three-layered structure, I'd recommend shortly mentioning the hypolimnion here.

**Author response:** Agreed

**7. Referee comment:** L23: If needed you could also exchange 'approaches' and 'methods' with 'algorithms' throughout the manuscript.

**Author response:** Agreed.

**8. Referee comment:** L28: The phrase 'complex water column structures' is a bit confusing here. Are you referring to cases when the three-layer paradigm is violated? Maybe rephrase to complex thermal water column structure?

**Author response:** Agreed.

**9. Referee comment:** L35: The phrase 'less problematic' is quite vague, do you mean 'introduces less bias'?

**Author response:** Good point, we would like to simplify the sentence to;

L35. *'While there is no prescribed rationale for selecting a particular method, the method which defined the epilimnion depth as the shallowest depth where the density was 0.1 kg m$^{-3}$ more than the surface density, may be particularly useful as a generic method.'*

**10. Referee comment:** L40: What do you mean by 'rapid gradients'? I'd recommend 'steep gradients'

**Author response:** Agreed.

**11. Referee comment:** L48-50: I'd recommend stressing these important statements more throughout the discussion at the end

**Author response:** Agreed. Please see the small edit below, in addition to updated discussion paragraph on L453.

L.48: '*The discretisation of these layers, however, is understood to be essentially theoretical, since micro-profile studies show that the conditions within layers are not uniform and exact cut-offs between layers do not necessarily exist (Imberger, 1985, Jonas et al., 2003, Tedford 2014, Kraemer et al., 2020).The definition of the epilimnion depth is thus inherently subjective, but has profound importance in limnology.*'

L453:'*Regardless of the method selected, however, all water temperature/density based methods are limited in their ability to indicate actual mixing processes. Our results using the lake modelled turbulence data demonstrated that even in a modelled environment, epilimnion depth estimates were inconsistent between the different methods and threshold values studied, and that turbulence based methods generally resulted in a shallower epilimnion depth estimate. These findings highlight the important but subtle difference between the layer detected by water density profiles (i.e. has been recently well-mixed and therefore has little resistance to further mixing due to the lack of density gradients), and the layer that is actively mixing, determined only through directly measured turbulence (Gray et al., 2020). Similarly, micro-profiling studies have shown that the actively mixing layer can be substantially shallower than the layer determined through water temperature profile data (McIntyre et al., 1993, Tedford et al., 2014). Micro-profile studies also demonstrate that within seemingly uniform layers there are micro-stratification layers, delineated by temperature differences as small as 0.02°C (Imberger, 1985, Shay and Gregg, 1986, MacIntyre, 1993; Jonas et al. 2003), which can be sufficient to isolate intermediate layers from atmospheric wind shear and cooling (Pernica et al., 2014). Although our results are not directly indicative of measured data, they demonstrate how even turbulence based methods are inherently arbitrary, as there is no objective threshold value (Monismith and Macintyre, 2009). Many of the ecological applications of the epilimnion depth have the underlying assumption that enough mixing is occurring in the epilimnion to keep the relevant organisms or particles suspended within the layer. Whether mixing is actually occurring, however, and to what extent, is not directly described by epilimnion depth estimations derived using water temperature or density profile data, and in fact, previous studies have found water density estimates of the epilimnion depth to be relatively poor indicators for the homogeneity of other ecological variables (Gray et al., 2020).*'

**12. Referee comment:** L74-75: Seems like sentencesL48-50 already explained that in reality there are no exact cut-offs, so how could there even be a consistent method used throughout limnology?

**Author response:** Interpretation of multi-lake studies or comparisons between separate studies, does require either consistent methods or an understanding of how/why the differences in methods will influence the results. We agree however that there cannot be an objective definition and therefore will change L74-75 to;

*L.74: Despite the ubiquity of the epilimnion depth, there is no consistent method used in limnology.*

**13. Referee comment:** L86: Is the vertical turbulence profile referring to a profile of turbulent eddy diffusivities? Here, the authors could also discuss field methods which measure turbulence in lakes, e.g. through velocity loggers. Or methods estimating diffusivities from water temperature profiles, e.g. gradient flux method by Heinz et al (1990) or heat budget method by Jassby and Powell (1975)

**Author response:** Yes, please see L86 rephrased below. Unfortunately, estimating vertical diffusivity from water temperature profiles as shown in Heinz et al., (1990) and Jassby and Powell (1975) can only estimate diffusivity below the epilimnion and below the photic zone. In addition, they can only estimate diffusivity over large temporal aggregates, depending on the accuracy of temperature sensors, and are also dependent on the (usually unknown) flux of heat with the sediment. Please also seeComment #11 for edits to discussion relating to turbulence.

*L85: 'Compared with long-term water temperature datasets, there are relatively few turbulent eddy diffusivity measurements in lakes, typically using micro-profiling methods conducted over over a small time period (e.g. Imberger, 1985, Tedford, 2014). Other methods of estimating vertical eddy diffusivity, from water temperature data for example, as in the Jassby and Powell (1975) heat-flux method are restricted to use below the epilimnion and photic zone. Therefore, epilimnion depth definitions based on actual turbulence measurements are uncommon. Vertical turbulence profiles, however, as well as water temperature profiles, are estimated by some hydrodynamic lake models (Goudsmit et al., 2002, Dong et al., 2019). Such modelled data, therefore, offers a tool for assessing commonly used water temperature/density based methods in comparison to turbulence based methods. '*

**14. Referee comment:** L116: I'd recommend moving the sentence "The lakes differ in many characteristics, [. . .]" to the beginning of the paragraph

**Author response:** Agreed.

**15. Referee comment:** 2.3 Simulated data: Tab. 1 suggests that the model was calibrated and validated. Which time periods were used? In this paragraph, the investigated fit criteria should also be mentioned. Also, in line 153, was the parameter of the minimum turbulent kinetic energy calibrated or, alternatively, which parameters affecting the min. turbulent kinetic energy were calibrated? This sentence is rather unclear.

**Author response:** Good point. We propose to update Section 2.3 with additional information on the calibration and validation process, in the text, and an additional Table in supplementary providing the model parameters and calibrated values, see below. The wnd_factor parameter is particularly important for the amount of turbulent kinetic energy available for mixing (Ayala et al. 2020).

L147 – 154: *'The Global Ocean Turbulence Model (GOTM), adapted for use in lakes, simulates small-scale turbulence and vertical mixing (Burchard et al,. 1999, Sachse et al., 2014, Moras et al., 2019, Ayala et al., in review) and was used to simulate daily profiles of water temperature (°C) and vertical eddy diffusivity (m-2 s -1) for Lake Erken and Lough Feeagh. A period of 4 years (1year spin-up followed by 3 years of calibration) was selected for calibration of GOTM, 2006-2009 for Lake Erken and 2008-2011 for Lough Feeagh. The model parameters that were calibrated were surface heat-flux factor (shf_factor), short-wave radiation factor (swr_factor), wind factor (wind_factor), minimum turbulent kinetic energy (k_min) and e-folding depth for visible fraction of light (g2) (See Apendix. Table S1.).'*

**Table S1.** Lake model parameters and calibrated values.

| Parameter | Lake Erken | Lough Feeagh |
|---|---|---|
| Shf_factor | 0.88 | 0.77 |
| Swr_factor | 0.98 | 0.93 |
| Wind_factor | 1.41 | 1.31 |
| K_min | 1.86e-6 | 3.48e-6 |
| G2 | 2.14 | 0.56 |

*'The validation period was 7 years (2010-2016) for Lake Erken and 4 years (2012-2015) for Lough Feeagh. For both calibration and validation, daily mean water temperatures were simulated when GOTM was forced using measured mean hourly. Model simulated profiles of mean daily water temperature were then compared to mean daily measured water temperature. During calibration the model was run approximately 5000 times to obtain a stable solution. Model performance was evaluated by comparing mean daily modelled and measured temperature profiles, the model efficiency coefficients used were percent relative error (PRE), root mean squared error (RMSE) and Nash-Sutcliffe efficiency (NSE) (Nash and Sutcliffe, 1970) (Table 2).'*

**Table 1.** Lake model performance evaluation, showing the percentage relative error (%), root mean squared error (°C), and Nash Sutcliffe efficiency, for Lough Feeagh (profiles = 1016, years = 5) and Lake Erken (profiles = 1449, years = 7).

| Statistic | Lough Feeagh | | Lake Erken | |
|---|---|---|---|---|
| | Calibration | Validation | Calibration | Validation |
| PRE (%) | -0.48 | 0.47 | -1.85 | 1.36 |
| RMSE (°C) | 0.67 | 1.18 | 0.53 | 0.55 |
| NSE | 0.97 | 0.92 | 0.98 | 0.97 |

**16. Referee comment:** 2.5 Analysis methods: This paragraph would benefit from sub-headings to make it easier for the reader to follow (similar structure as the Results would be beneficial).

**Author response:** We agree and can follow the same structure as the results.

**17. Referee comment:** L208: Here the colors are mentioned but the figure is not referenced. If you do not want to cross-reference the figure yet, maybe just write about 'color-coding'.

**Author response:** Agreed.

**18. Referee comment:** L232: 'logical and numerical schemes' is unclear to me. What do you mean by logical scheme here?

**Author response:** Good point, this is over-complicated language. We rephrase to:

L.231 'depending on the calculations used, may regularly encroach on the metalimnion (Lorbacher et al., 2006).'

**19. Referee comment:** L248: Is there a reason why the sensor deployment sensitivity was not tested on monitored data (e.g. by removing some loggers)?

**Author response:** Yes, we did that way also. However since Lough Feeagh was already quite coarse in vertical resolution, it did not seem like a fair comparison between the two lakes. The way we did it allowed us to see how the coarse resolution in Lough Feeagh could influence comparison between Erken, thus demonstrating that comparing lakes with different resolution will not be an equal comparison.

**20. Referee comment:** 3.2 Comparison between water density based methods: In my opinion, this paragraph is a crucial finding of the study as it discusses how deviations from the three-layer paradigm affected the results of the algorithms. Can you state how many profiles/observations

points were either a) (classical paradigm), b) (multiple pycnoclines) or c) (weakly stratified)? Were most profiles following the three-layer structure or is the complex case b) dominating?

**Author response:** The difficulty with doing this is that there is no established method for classifying multiple pycnoclines or weak stratification. Thus, the results would be dependent on whichever arbitrary thresholds we used, and it is this dependence on arbitrary thresholds that we are highlighting in the study. Hence we do not think it would be appropriate within this paper. The topic is an interesting one though, and we think, worthy of a future study. However, we will dedicate a paragraph in the discussion to this consideration, which also addresses Comment #4 and Comment #11.

**21. Referee comment:** L301: Here it would be beneficial for the reader to state the mean epilimnion depth per lake plus the variance and the quantiles.

**Author response:** Agreed, see additional line below including the standard error and the inter-quartile range.

L301: '*The mean epilimnion depth estimate for all observed data, calculated with methods M1-M4 and all thresholds was -28.1 m (SEM =0.6 m, inter-quartile range = 19.0 m) for Lough Feeagh and -11.0 m (SEM = 0.1 and IQR = 2.3 m) for Lake Erken.*'

**22. Referee comment:** L330: Shouldn't it be '[. . .] epilimnion depth was identified at a depth above [. . .]' instead of "greater"? As everything is referenced to the surface, wouldn't greater mean deeper?

**Author response:** Agreed, thank you.

**23. Referee comment:** L332: It seems there's a word missing here

**Author response:** Agreed,

L332: '*For M4, the percentage of stratified days remained static regardless of the threshold value, because the epilimnion depth was detected for all profiles where the water column temperature range was more than 1 °C, regardless of the threshold used.*'

**24. Referee comment:** L336: Do you mean 0.025 kg/m3/m instead of 0.25 kg/m3/m which would be higher than the maximum investigated threshold?

**Author response:**. Thank you, updated to 0.025 kg/m3/m

**25. Referee comment:** L360: The phrase "[. . .] M2 had typically a higher threshold range than M3 [. . .]" confuses me. Do you mean that the range between the thresholds was higher?

**Author response:** Yes, rephrased:

L360: '*M2 had typically a greater range in epilimnion depth estimates than M3, when calculated with the same range of thresholds.*'

**26. Referee comment:** 3.4 Sensitivity of epilimnion depth: Can you test if the differences between the means were significant?

**Author response:** We consider that assessing whether the differences between the epilimnion depth estimates derived using high/low vertical resolution water temperature datasets were statistically significant would not be appropriate. This is because the lower resolution water temperature dataset is nested within the higher resolution dataset, and therefore the estimated MLD values based on these data would not be independent. Moreover, we know that these two datasets do differ, as we have induced that difference intentionally by filtering the higher resolution modelled data to obtain the lower resolution dataset. Our aim here was to quantify the change in epilimnion depth estimates for each method, accounting for when data might be available at high resolution intervals (our modelled 0.5 m data) or available using a deployment strategy similar to that currently employed in Lough Feeagh (mean of one sensor per 3 m). The result demonstrated how M1 in particular tended to be less sensitive to the vertical resolution of the input data than other methods, since the differences between the mean high and lower resolution data derived epilimnion depth estimates were smallest. In order to make the spread of the data more apparent to the reader, we have added the standard error for each of the means in Table 4 (see below). This will give a measure of the variability around the mean and allow a more informed assessment of the difference between the derived MLD values. For consistency, the standard error will also be added to Table 2, the table showing the observed results.

**Table 4.** Mean Apr-Oct epilimnion depth estimates (m) derived using high resolution and low resolution modelled water temperature data with standard error of the mean in brackets, and the difference calculated between the high resolution and low resolution estimate (m), for Lough Feeagh and Lake Erken.

| Method | Lough Feeagh | | | Lake Erken | | |
|---|---|---|---|---|---|---|
| | High resolution mean (m) | Low resolution mean (m) | Difference in mean epilimnion depth (m) | High resolution mean (m) | Low resolution mean (m) | Difference in mean epilimnion depth (m) |
| M1 | -22.1 (0.7) | -22.2 (0.7) | 0.1 | -10.9 (0.1) | -10.9 (0.1) | 0.0 |
| M2 | -31.7 (1.1) | -34.9 (1.2) | 3.2 | -11.9 (0.2) | -13.1 (0.2) | 1.2 |
| M3 | -32.0 (1.1) | -35.2 (1.1) | 3.2 | -12.1 (0.2) | -13.1 (0.2) | 1.0 |
| M4 | -22.1 (0.6) | -22.6 (0.6) | 0.5 | -11.3 (0.1) | -11.5 (0.1) | 0.2 |

**Table 2.** Summary of statistics for each method, showing the mean (m) and standard error of mean in brackets, minimum (i.e. shallowest estimate) (m), maximum (i.e. deepest estimate) (m) and the interquartile range (m) of the April-October epilimnion depth estimates (summarised from the results shown in Fig.6a), and the mean Pearson's correlation coefficient (r) for each method, representing the mean correlation for all possible combinations between threshold values., for Lough Feeagh and Lake Erken.

| Method | Lough Feeagh | | | | | Lake Erken | | | | |
|---|---|---|---|---|---|---|---|---|---|---|
| | Mean (m) | Min (m) | Max (m) | IQR (m) | r | Mean (m) | Min (m) | Max (m) | IQR (m) | r |
| M1 | -19.0 (0.8) | -4.6 | -25.4 | 7.3 | 0.77 | -10.0 (0.2) | -7.8 | -11.2 | 1.3 | 0.92 |
| M2 | -35.9 (0.9) | -19.7 | -41.4 | 6.5 | 0.48 | -11.3 (0.2) | -8.4 | -12.9 | 1.7 | 0.78 |
| M3 | -36.5 (0.8) | -22.4 | -41.5 | 6.1 | 0.49 | -11.8 (0.2) | -10.0 | -13.0 | 1.2 | 0.82 |
| M4 | -21.1 (0.5) | -19.7 | -21.5 | 0.3 | 1.00 | -11.9 (0.1) | -10.1 | -11.3 | 0.5 | 0.99 |

**27. Referee comment:** L405: The phrase 'particularly distinct systematic difference' seems a bit excessive

**Author response:** Good point, rephrased to;

L405: *'large systematic difference'*

**28. Referee comment:** L406: Is '[. . .] was equivalent to using different threshold values [. . .]' referring to density threshold values?

**Author response:**  Agreed, rephrased to;

L406:*'Due to the non-linear relationship between water density and temperature, the use of water temperature was equivalent to using different density threshold values throughout the year, resulting in a distinct shift in the stratification period.'*

**29. Referee comment:** L424: Do you actually mean 'acceleration of epilimnion deepening' instead of shallowing here? Shouldn't the epilimnion deepen relative to the surface height during stratification onset?

**Author response:** Good point, this is not clear, we have rephrased to,

L.423 *'Alternatively, water temperature-based estimates typically resulted in earlier stratification, which could indicate a longer duration of phytoplankton in a shallower epilimnion.'*

**30. Referee comment:** L429: But these methods are detecting a layer that is specifically not isothermal relative to themselves, right?

**Author response:** Correct. To make this clearer we have rephrased.

L.429, *'An important difference was also found between 1) methods detecting the layer that is isothermal relative to the surface and 2) methods detecting the point that is isothermal relative to the steep gradients of the metalimnion.'*

**31. Referee comment:** L448: The phrase '[. . .] not be suitable for use with water density metric [. . .]' is unclear to me. Which water density metric are you talking about?

**Author response:** We are pointing out that use of water density metrics (e.g. rLakeanalyzer meta.tops) with a water temperature based stratification definitions (e.g. <1°C) is incompatible, and could lead to a seasonal bias. This is based on the results shown in Figure 3. We have rephrased this sentence as below.

*L448. 'The results suggest that use of water density metrics, such as epilimnion depth estimates, in combination with traditional water temperature based definitions of stratification, are incompatible, given the non-linear relationship between temperature and density'*

**32. Referee comment:** L453-466: A figure showing these results would be beneficial for the reader, or how M5 compares to the other models.

**Author response:** We propose to add a time series figure to the Supplementary Material showing the profile-based methods (M1-M4) and the turbulence method (M5), using all thresholds, for one year of data, as below.

[Figure]

**Figure S1:** Daily epilimnion depth estimates using modelled data for 2016 from Lough Feeagh and Lake Erken, showing estimates from all profile based epilimnion depth methods, including M1, the absolute difference from the surface method (**a**), M2, the gradient from the surface method (**b**), M3, the gradient from the pycnocline method (**c**) and M4, the rLakeAnalyzer method (**d**), as well as M5, the modelled turbulence based method (**e**), calculated using the full range of thresholds, and for each lake.

**33. Referee comment:** L465: Are profile data here referring to water temperature profile data?

**Author response:** Yes, will be changed to read 'water temperature profile data' explicitly.

**34. Referee comment:** L515: I'd recommend not to write about 'problematic' here, maybe 'less bias', 'conservative assumption'?

**Author response:** L.515 '*less sensitive to vertical sensor resolution than the other methods*'

**35. Referee comment:** Table 3: Could you exchange the table with either a correlation matrix or correlation plot?

**Author response:** We propose to add a correlation matrix for each lake in the supplementary material (see Tables below) since they are very large tables, while keeping Table 3. as it is a useful summary for the main text.

**Table S2:** Correlation matrix between all methods and all threshold combinations for Lough Feeagh and Lake Erken.

**Lough Feeagh**

| | | Method 1 | | | | | | | | Method 2 | | | | | | | | Method 3 | | | | | | | | Method 4 | | | | | | | |
|---|---|---|---|---|---|---|---|---|---|---|---|---|---|---|---|---|---|---|---|---|---|---|---|---|---|---|---|---|---|---|---|---|---|
| | | 0.03 | 0.05 | 0.08 | 0.1 | 0.13 | 0.15 | 0.18 | 0.2 | 0.03 | 0.05 | 0.08 | 0.1 | 0.13 | 0.15 | 0.18 | 0.2 | 0.03 | 0.05 | 0.08 | 0.1 | 0.13 | 0.15 | 0.18 | 0.2 | 0.03 | 0.05 | 0.08 | 0.1 | 0.13 | 0.15 | 0.18 | 0.2 |
| Method 1 | 0.025 | | | | | | | | | | | | | | | | | | | | | | | | | | | | | | | | |
| | 0.05 | .59 | | | | | | | | | | | | | | | | | | | | | | | | | | | | | | | |
| | 0.075 | .46 | .82 | | | | | | | | | | | | | | | | | | | | | | | | | | | | | | |
| | 0.1 | .42 | .76 | .95 | | | | | | | | | | | | | | | | | | | | | | | | | | | | | |
| | 0.125 | .41 | .73 | .91 | .98 | | | | | | | | | | | | | | | | | | | | | | | | | | | | |
| | 0.15 | .39 | .71 | .89 | .95 | .99 | | | | | | | | | | | | | | | | | | | | | | | | | | | |
| | 0.175 | .38 | .69 | .86 | .93 | .97 | .99 | | | | | | | | | | | | | | | | | | | | | | | | | | |
| | 0.2 | .37 | .67 | .85 | .91 | .95 | .97 | .99 | | | | | | | | | | | | | | | | | | | | | | | | | |
| Method 2 | 0.025 | .41 | .68 | .73 | .74 | .76 | .78 | .79 | .79 | | | | | | | | | | | | | | | | | | | | | | | | |
| | 0.05 | .26 | .48 | .61 | .65 | .68 | .70 | .72 | .74 | .64 | | | | | | | | | | | | | | | | | | | | | | | |
| | 0.075 | .16 | .30 | .39 | .43 | .46 | .49 | .51 | .53 | .40 | .67 | | | | | | | | | | | | | | | | | | | | | | |
| | 0.1 | .11 | .22 | .29 | .33 | .36 | .38 | .40 | .42 | .30 | .51 | .77 | | | | | | | | | | | | | | | | | | | | | |
| | 0.125 | .08 | .16 | .22 | .25 | .27 | .29 | .30 | .32 | .22 | .38 | .58 | .75 | | | | | | | | | | | | | | | | | | | | |
| | 0.15 | .07 | .13 | .17 | .21 | .23 | .24 | .25 | .26 | .18 | .31 | .46 | .60 | .79 | | | | | | | | | | | | | | | | | | | |
| | 0.175 | .06 | .11 | .15 | .17 | .19 | .21 | .22 | .23 | .15 | .26 | .38 | .49 | .66 | .83 | | | | | | | | | | | | | | | | | | |
| | 0.2 | .05 | .09 | .13 | .15 | .17 | .18 | .19 | .19 | .13 | .22 | .33 | .42 | .56 | .71 | .86 | | | | | | | | | | | | | | | | | |
| Method 3 | 0.025 | .39 | .64 | .68 | .71 | .74 | .77 | .78 | .79 | .94 | .61 | .43 | .34 | .26 | .22 | .19 | .16 | | | | | | | | | | | | | | | | |
| | 0.05 | .25 | .46 | .58 | .63 | .66 | .68 | .70 | .72 | .61 | .95 | .71 | .55 | .42 | .34 | .28 | .24 | .61 | | | | | | | | | | | | | | | |
| | 0.075 | .15 | .28 | .37 | .42 | .45 | .48 | .50 | .52 | .38 | .64 | .97 | .80 | .60 | .48 | .40 | .34 | .43 | .69 | | | | | | | | | | | | | | |
| | 0.1 | .11 | .21 | .27 | .31 | .34 | .36 | .38 | .40 | .29 | .49 | .73 | .95 | .79 | .63 | .53 | .45 | .33 | .53 | .76 | | | | | | | | | | | | | |
| | 0.125 | .08 | .15 | .20 | .23 | .26 | .27 | .29 | .30 | .21 | .36 | .54 | .70 | .94 | .85 | .71 | .61 | .25 | .39 | .56 | .74 | | | | | | | | | | | | |
| | 0.15 | .07 | .12 | .16 | .20 | .21 | .23 | .24 | .25 | .17 | .29 | .43 | .56 | .75 | .94 | .88 | .75 | .21 | .32 | .45 | .60 | .81 | | | | | | | | | | | |
| | 0.175 | .06 | .11 | .14 | .17 | .18 | .20 | .21 | .22 | .14 | .24 | .36 | .47 | .63 | .79 | .95 | .90 | .18 | .27 | .38 | .50 | .68 | .84 | | | | | | | | | | |
| | 0.2 | .05 | .09 | .12 | .14 | .16 | .17 | .18 | .18 | .12 | .21 | .31 | .40 | .54 | .68 | .81 | .95 | .15 | .23 | .32 | .43 | .57 | .71 | .85 | | | | | | | | | |
| Method 4 | 0.025 | .40 | .70 | .82 | .81 | .78 | .75 | .72 | .70 | .58 | .50 | .36 | .29 | .22 | .19 | .17 | .15 | .62 | .50 | .36 | .28 | .22 | .19 | .16 | .14 | | | | | | | | |
| | 0.05 | .39 | .68 | .79 | .78 | .75 | .72 | .69 | .68 | .55 | .48 | .36 | .29 | .23 | .20 | .18 | .15 | .59 | .49 | .36 | .29 | .23 | .20 | .17 | .15 | 1.00 | | | | | | | |
| | 0.075 | .39 | .67 | .78 | .77 | .74 | .71 | .68 | .66 | .53 | .46 | .34 | .29 | .23 | .21 | .18 | .16 | .58 | .47 | .35 | .28 | .23 | .20 | .17 | .15 | .99 | 1.00 | | | | | | |
| | 0.1 | .39 | .67 | .78 | .77 | .74 | .70 | .67 | .65 | .53 | .45 | .33 | .27 | .23 | .21 | .18 | .16 | .58 | .46 | .33 | .27 | .22 | .20 | .17 | .15 | .99 | 1.00 | 1.00 | | | | | |
| | 0.125 | .39 | .67 | .78 | .77 | .73 | .70 | .67 | .65 | .53 | .44 | .32 | .26 | .22 | .20 | .18 | .15 | .58 | .45 | .32 | .26 | .22 | .19 | .17 | .15 | .99 | 1.00 | 1.00 | 1.00 | | | | |
| | 0.15 | .39 | .67 | .78 | .77 | .73 | .70 | .67 | .65 | .53 | .44 | .32 | .26 | .22 | .20 | .18 | .15 | .58 | .45 | .32 | .25 | .21 | .19 | .17 | .15 | .99 | 1.00 | 1.00 | 1.00 | 1.00 | | | |
| | 0.175 | .39 | .67 | .78 | .77 | .73 | .70 | .67 | .65 | .53 | .44 | .31 | .26 | .21 | .19 | .17 | .15 | .58 | .45 | .32 | .25 | .21 | .19 | .17 | .14 | .99 | 1.00 | 1.00 | 1.00 | 1.00 | 1.00 | | |
| | 0.2 | .39 | .67 | .78 | .77 | .73 | .70 | .67 | .65 | .53 | .44 | .31 | .26 | .21 | .19 | .17 | .15 | .58 | .45 | .32 | .25 | .21 | .18 | .16 | .14 | .99 | 1.00 | 1.00 | 1.00 | 1.00 | 1.00 | 1.00 | |

**Lake Erken**

| | | Method 1 | | | | | | | | Method 2 | | | | | | | | Method 3 | | | | | | | | Method 4 | | | | | | | |
|---|---|---|---|---|---|---|---|---|---|---|---|---|---|---|---|---|---|---|---|---|---|---|---|---|---|---|---|---|---|---|---|---|---|
| | | 0.03 | 0.05 | 0.08 | 0.1 | 0.13 | 0.15 | 0.18 | 0.2 | 0.03 | 0.05 | 0.08 | 0.1 | 0.13 | 0.15 | 0.18 | 0.2 | 0.03 | 0.05 | 0.08 | 0.1 | 0.13 | 0.15 | 0.18 | 0.2 | 0.03 | 0.05 | 0.08 | 0.1 | 0.13 | 0.15 | 0.18 | 0.2 |
| Method 1 | 0.025 | | | | | | | | | | | | | | | | | | | | | | | | | | | | | | | | |
| | 0.05 | .94 | | | | | | | | | | | | | | | | | | | | | | | | | | | | | | | |
| | 0.075 | .89 | .97 | | | | | | | | | | | | | | | | | | | | | | | | | | | | | | |
| | 0.1 | .85 | .94 | .98 | | | | | | | | | | | | | | | | | | | | | | | | | | | | | |
| | 0.125 | .82 | .91 | .96 | .99 | | | | | | | | | | | | | | | | | | | | | | | | | | | | |
| | 0.15 | .79 | .89 | .94 | .97 | .99 | | | | | | | | | | | | | | | | | | | | | | | | | | | |
| | 0.175 | .77 | .87 | .92 | .96 | .98 | 1.00 | | | | | | | | | | | | | | | | | | | | | | | | | | |
| | 0.2 | .75 | .85 | .90 | .94 | .97 | .99 | 1.00 | | | | | | | | | | | | | | | | | | | | | | | | | |
| Method 2 | 0.025 | .85 | .92 | .92 | .89 | .87 | .85 | .83 | .81 | | | | | | | | | | | | | | | | | | | | | | | | |
| | 0.05 | .78 | .87 | .92 | .94 | .95 | .95 | .95 | .94 | .85 | | | | | | | | | | | | | | | | | | | | | | | |
| | 0.075 | .70 | .78 | .84 | .87 | .90 | .92 | .92 | .93 | .75 | .90 | | | | | | | | | | | | | | | | | | | | | | |
| | 0.1 | .64 | .73 | .78 | .82 | .85 | .87 | .88 | .89 | .70 | .84 | .94 | | | | | | | | | | | | | | | | | | | | | |
| | 0.125 | .59 | .67 | .72 | .75 | .78 | .81 | .83 | .85 | .64 | .77 | .86 | .93 | | | | | | | | | | | | | | | | | | | | |
| | 0.15 | .55 | .62 | .66 | .70 | .73 | .76 | .78 | .80 | .59 | .71 | .80 | .87 | .94 | | | | | | | | | | | | | | | | | | | |
| | 0.175 | .50 | .57 | .61 | .64 | .67 | .69 | .72 | .74 | .54 | .65 | .74 | .80 | .87 | .93 | | | | | | | | | | | | | | | | | | |
| | 0.2 | .45 | .52 | .55 | .58 | .61 | .63 | .66 | .68 | .49 | .59 | .67 | .73 | .80 | .86 | .93 | | | | | | | | | | | | | | | | | |
| Method 3 | 0.025 | .75 | .83 | .86 | .87 | .88 | .88 | .88 | .88 | .84 | .86 | .82 | .79 | .73 | .69 | .64 | .58 | | | | | | | | | | | | | | | | |
| | 0.05 | .72 | .81 | .86 | .89 | .91 | .92 | .93 | .93 | .78 | .92 | .89 | .86 | .81 | .77 | .72 | .66 | .91 | | | | | | | | | | | | | | | |
| | 0.075 | .66 | .74 | .79 | .83 | .86 | .88 | .90 | .91 | .71 | .85 | .94 | .92 | .88 | .84 | .78 | .72 | .83 | .92 | | | | | | | | | | | | | | |
| | 0.1 | .62 | .70 | .75 | .78 | .82 | .84 | .86 | .87 | .67 | .80 | .90 | .96 | .92 | .88 | .82 | .77 | .79 | .88 | .95 | | | | | | | | | | | | | |
| | 0.125 | .58 | .65 | .70 | .73 | .76 | .79 | .81 | .83 | .63 | .75 | .84 | .90 | .97 | .94 | .88 | .82 | .73 | .82 | .89 | .94 | | | | | | | | | | | | |
| | 0.15 | .53 | .61 | .65 | .68 | .71 | .73 | .76 | .78 | .58 | .69 | .78 | .84 | .91 | .97 | .93 | .87 | .68 | .76 | .84 | .88 | .94 | | | | | | | | | | | |
| | 0.175 | .49 | .56 | .59 | .63 | .65 | .68 | .70 | .72 | .53 | .64 | .72 | .78 | .85 | .91 | .98 | .94 | .63 | .71 | .78 | .82 | .88 | .93 | | | | | | | | | | |
| | 0.2 | .45 | .51 | .54 | .57 | .60 | .62 | .64 | .66 | .48 | .58 | .66 | .72 | .78 | .84 | .91 | .98 | .57 | .65 | .72 | .76 | .81 | .87 | .93 | | | | | | | | | |
| Method 4 | 0.025 | .78 | .83 | .86 | .88 | .88 | .88 | .88 | .88 | .77 | .84 | .82 | .79 | .74 | .70 | .65 | .59 | .90 | .90 | .84 | .80 | .74 | .69 | .64 | .58 | | | | | | | | |
| | 0.05 | .76 | .81 | .83 | .85 | .86 | .86 | .86 | .86 | .74 | .82 | .81 | .79 | .75 | .71 | .66 | .61 | .88 | .90 | .85 | .81 | .76 | .71 | .65 | .60 | .99 | | | | | | | |
| | 0.075 | .74 | .79 | .82 | .83 | .84 | .84 | .84 | .84 | .72 | .80 | .79 | .78 | .74 | .71 | .66 | .62 | .86 | .88 | .85 | .81 | .76 | .71 | .66 | .61 | .98 | .99 | | | | | | |
| | 0.1 | .73 | .78 | .81 | .82 | .83 | .83 | .83 | .83 | .71 | .79 | .78 | .76 | .73 | .70 | .66 | .62 | .85 | .87 | .83 | .80 | .75 | .71 | .66 | .61 | .97 | .99 | 1.00 | | | | | |
| | 0.125 | .73 | .77 | .80 | .81 | .82 | .82 | .82 | .82 | .70 | .78 | .77 | .75 | .72 | .70 | .66 | .61 | .84 | .86 | .82 | .79 | .74 | .70 | .66 | .61 | .96 | .98 | .99 | 1.00 | | | | |
| | 0.15 | .72 | .77 | .79 | .81 | .81 | .81 | .81 | .81 | .69 | .77 | .75 | .74 | .71 | .69 | .65 | .61 | .84 | .86 | .81 | .78 | .73 | .69 | .65 | .61 | .96 | .98 | .99 | 1.00 | 1.00 | | | |
| | 0.175 | .72 | .76 | .79 | .80 | .81 | .81 | .81 | .80 | .69 | .76 | .75 | .73 | .70 | .67 | .64 | .60 | .83 | .85 | .80 | .77 | .72 | .68 | .64 | .60 | .95 | .98 | .99 | .99 | 1.00 | 1.00 | | |
| | 0.2 | .71 | .76 | .78 | .80 | .80 | .80 | .80 | .80 | .69 | .76 | .74 | .72 | .69 | .67 | .63 | .59 | .83 | .85 | .80 | .76 | .71 | .67 | .63 | .59 | .95 | .98 | .99 | .99 | 1.00 | 1.00 | 1.00 | |

**36. Referee comment:** Fig. 2 is great, I like it a lot! It makes the whole study easier to understand.

**Author response:** Thanks!

**37. Referee comment:** Fig 3.: Why is there a thin blue shaded area below the red shade in Lake Erken?

**Author response:** That is a graphical issue and will be fixed in the revised version.

**Additional References**

Heinz, G., Ilmberger, J. and Schimmele, M., 1990. Vertical mixing in Überlinger See, western part of Lake Constance. Aquatic sciences, 52(3), pp.256-268.

Jassby, A. and Powell, T., 1975. Vertical patterns of eddy diffusion during stratification in Castle Lake, California 1. Limnology and oceanography, 20(4), pp.530-543.

Marcé, R., George, G., Buscarinu, P., Deidda, M., Dunalska, J., de Eyto, E., Flaim, G., Grossart, H.P., Istvanovics, V., Lenhardt, M. and Moreno-Ostos, E., 2016. Automatic high frequency monitoring for improved lake and reservoir management. Environmental Science & Technology, 50(20), pp.10780-10794.

Pernica, P., Wells, M.G. and MacIntyre, S., 2014. Persistent weak thermal stratification inhibits mixing in the epilimnion of north-temperate Lake Opeongo, Canada. *Aquatic sciences*, *76*(2), pp.187-201.

Shay, T.J. and Gregg, M.C., 1986. Convectively driven turbulent mixing in the upper ocean. *Journal of Physical Oceanography*, *16*(11), pp.1777-1798.

Tedford, E.W., MacIntyre, S., Miller, S.D. and Czikowsky, M.J., 2014. Similarity scaling of turbulence in a temperate lake during fall cooling. *Journal of Geophysical Research: Oceans*, *119*(8), pp.4689-4713.